# Taming Variability: Randomized and Bootstrapped Conformal Risk Control for LLMs

## Abstract

We transform the randomness of LLMs into precise assurances using an actuator at the API interface that applies a user-defined risk constraint in finite samples via Conformal Risk Control (CRC). This label-free and model-agnostic actuator manages ship/abstain/regenerate/escalate actions based solely on a scalar score from opaque outputs. We enhance CRC's computational efficiency and robustness through Batched Bootstrap CRC (BB-CRC) and Randomized Batched Weighted-Average CRC (RBWA-CRC), reducing calibration calls and stabilizing thresholds while maintaining statistical validity. Additionally, we present a semantic quantification method grounded in gram matrix geometry, resulting in interpretable signal and metric design. Together these pieces deliver principled randomness control for LLM hallucination mitigation and LLM-as-judge reliability. Our framework is assessed using four datasets, demonstrating its efficacy in enhancing factual accuracy and measuring LLM-as-judge performance, yielding a simplified and computationally efficient control layer that converts variability into statistical validity.

## 1 Introduction

Recently developed large language models (LLMs) function as stochastic *black boxes*, limiting common user access to logits or internal processes during deployment. Key issues include fabricated information, hallucination, prompt-injection vulnerabilities, attacks, and inconsistent evaluations when LLMs evaluate their own outputs, resulting in compromised outputs. These problems undermine large-scale reliability and safety due to a lack of explicit, analytical, and scalable uncertainty control (Aljamaan et al., 2024; Alizadeh et al., 2025; Wang et al., 2025b; Zheng et al., 2023; Shi et al., 2025; Chen et al., 2024). A *flexible, vendor-independent control framework* that accounts for computational context and converts variability into *probabilistic guarantees* essential for practitioners is missing.

We present a *Conformal Actuator (CA) framework*, a single, monotone gate that directs actions (ship, abstain, regenerate, escalate) via a scalar, label-free score derived from outputs of black-box models. Calibrated once via *Conformal Risk Control (CRC)*, the CA enforces a pre-specified risk budget with finite-sample guarantees. At the API boundary, we create two efficient and analytically tractable calibrators—**Batched-Bootstrap CRC (BB-CRC)** and **Randomized Batched Weighted-Average CRC (RBWA-CRC)**—that preserve probabilistic validity in finite samples while reducing calls and smoothing calibration, improving statistical efficiency and robustness. Beyond gating, we add a *quantification layer* that *quantifies and bounds* any offline-flagged risk *by a geometrically principled centered Gram-matrix metric*. Crucially, calibration folds ground-truth information into this *cheap, API-only* Gram signal, so the deployed score remains label-free yet inherits statistical guarantees.

We focus on two deployment challenges most significantly impacted by randomness and uncertainty.**(i) Hallucination control.** Can we ship only answers that are likely factual, while bounding the "acted-while-unfactual" exposure? **(ii) LLM-as-Judge reliability.** In utilizing an LLM evaluator for reviewing outputs from other LLMs, to what extent can we rely on this mechanism given our foundational mistrust of LLM-generated responses? A runtime process that is label-free, operationally simple, and statistically sound is necessary in both instances.

We formalize the CA and its guarantees under CRC, present the compute-aware calibrators (**BB-CRC**, **RBWA-CRC**), and evaluate the full system on open-domain QA. Across benchmarks, CRC-calibrated routing achieves consistent *factuality lifting* at the target budget, while the Gram-geometry score—cheap to compute at inference and label-free—delivers the most uniform gains. **Overall, we provide an end-to-end, verifiable deployment pipeline that is (i) label-free at inference, (ii) compute-aware in calibration, and (iii) statistically valid in finite samples—combining a efficient actuator with a geometry-based quantification layer for practical, auditable control.**

## 2 RELATED WORK

*Conformal prediction (CP)* turns arbitrary model scores into uncertainty sets with *distribution-free, finite-sample* guarantees under minimal assumptions, spanning classical tutorials and modern variants such as split CP for regression and conformalized quantile regression (Vovk et al., 2005; Shafer & Vovk, 2007; Lei et al., 2017; Romano et al., 2019). Robustness beyond exchangeability and auditing under shift have been actively studied (Oliveira et al., 2024; Prinster et al., 2022; Tibshirani et al., 2020), while domain surveys and task-specific adaptations cover NLP and LLM-style outputs (Campos et al., 2024; Quach et al., 2024; Mohri & Hashimoto, 2024). *Conformal Risk Control* (CRC) extend CP from coverage to *expected-loss control* for bounded, monotone losses, yielding data-dependent thresholds with finite-sample guarantees that transfer to deployment (Bates et al., 2021; Angelopoulos & Bates, 2022; Angelopoulos et al., 2025). Methodological extensions include cross-validated calibration and anytime-valid/sequential control (Cohen et al., 2024; Xu et al., 2024). In LLM pipelines, CRC has been used for tail-risk alignment, property alignment, metric calibration, and multi-objective routing/cascades directly at the API layer (Overman et al., 2024; Overman & Bayati, 2025; Chen et al., 2025; Gomes et al., 2025). Yadkori et al. (2024) employs CRC to reduce hallucinations by implementing a conformal-abstention strategy with standard scoring and calibrator.

Without white-box access, one can embed a small batch of outputs, form a Gram matrix (Schölkopf et al., 1998), and summarize *consensus* or *uncertainty* via functionals; this connects to representation-similarity (CKA) and recent semantic-space uncertainty measures including semantic entropy, kernel language entropy, and log-det (semantic volume) scores (Kornblith et al., 2019; Farquhar et al., 2024; Kossen et al., 2024; Nikitin et al., 2024; Li et al., 2025). Signals based on geometry assist in hallucination management, self-consistency, and consensus analysis.(Liu et al., 2023b; Manakul et al., 2023; Wang et al., 2025a).

## 3 GRAM GEOMETRY FOR BLACK-BOX LLM RESPONSES: SEMANTIC SUFFICIENCY AND QUANTIFICATION

We require a metric layer that is mathematically stable and depends only on black-box LLM outputs. Building on *self-consistency*—which aggregates multiple reasoning paths to improve reliability (Wang et al., 2023)—and the *Semantic Volume* view that links uncertainty to embedding dispersion via log determinants (Li et al., 2025), we work directly in Gram space. This yields permutation invariance, numerical stability, and an outputs-only interface. On this basis, we design a novel *response-level* Gram metric that is cheap to compute and readily deployable for safety control, providing a label-free signal that integrates seamlessly with our conformal actuator.

Let $v_i = \psi(y_i) \in \mathbb{R}^d$ be unit-norm embeddings of $n$ i.i.d. responses in a small queue, stack $V \in \mathbb{R}^{n \times d}$, define $G := VV^\top$, center with $H := I_n - \frac{1}{n}\mathbf{1}\mathbf{1}^\top$, and write $\tilde{G} := HGH$. This yields two deployable capabilities: *(A) semantic sufficiency* for batch decisions via leading subspaces, and *(B) per-item quantification* via a one-dimensional, intrinsic uncertainty score—both label-free online.

**Semantic sufficiency: decisions live in leading Gram subspaces.** We compare the leading rank-$r$ projector of a test batch to class prototypes. For each class $k$, average calibration Grams to a surrogate $S_k$ (eigengap $\gamma_k > 0$), extract its top-$r$ projector $P_k$, and build prototype projectors $\hat{P}_k$ from held-out data. For a test batch form $\hat{P}$ from the top-$r$ eigenvectors of $\tilde{G}$. Decide via the *spectral–overlap* rule

$$\hat{k} = \arg\max_k \langle \hat{P}, \hat{P}_k \rangle_F. \tag{3.1}$$

If the centered Gram of the batch concentrates near its true class mean ($\|\tilde{G} - S_{k^\star}\|_{\mathrm{op}} \leq \varepsilon_n$) and prototypes are separated, then $\hat{P}$ is close to $P_{k^\star}$ and the overlap rule equation A.1 returns the correct class with a positive margin. Formally:

**Theorem 3.1** (Semantic sufficiency of Gram projectors). *Let the true class be $k^\star$ and suppose $\|\tilde{G} - S_{k^\star}\|_{\mathrm{op}} \leq \varepsilon_n$. Define the* prototype separation $\Delta_P := \min_{j \neq \ell} \|\hat{P}_j - \hat{P}_\ell\|_F$ *and the* prototype error $\delta_{\mathrm{proto}} := \|\hat{P}_{k^\star} - P_{k^\star}\|_F$. *If*

$$\frac{2\sqrt{r}}{\gamma_{k^\star}} \, \varepsilon_n \; + \; \delta_{\mathrm{proto}} \; < \; \frac{1}{4} \, \Delta_P, \tag{3.2}$$

*then the spectral–overlap rule equation A.1 selects $\hat{k} = k^\star$ with margin*

$$m \; := \; \langle \hat{P}, \hat{P}_{k^\star} \rangle_F \; - \; \max_{j \neq k^\star} \langle \hat{P}, \hat{P}_j \rangle_F \; \geq \; \frac{\Delta_P^2}{4} \; > \; 0 \,. \tag{3.3}$$

The Davis–Kahan projector bound and the margin argument are given in the appendix; all lemmas are deferred to Appendix §A.1.[1]

**Quantification: a one-dimensional, intrinsic energy scale.** Per-item uncertainty is quantified using the *interaction energy*

$$e(i; G) := \|G_{:,i}\|_2 = \|V \, v_i\|_2.$$

With unit-norm embeddings, $e(i; G)^2 = \sum_{j=1}^{n} \cos^2 \theta_{ij}$, so large $e$ indicates batch consensus (alignment or anti-alignment both count) and small $e$ indicates novelty. The scale is *intrinsic*: $1 \leq e(i; G) \leq \sqrt{n}$, hence the normalized score $E(i) := e(i; G)/\sqrt{n} \in [0, 1]$ is a label-free, permutation-invariant policy signal compatible with CRC.

We instantiate the *Gram–Projector Spectral-Overlap (GPSO)* decision rule and verify that a simple L2/no-center pipeline achieves a best macro accuracy of **0.958** on a factual vs. unfactual QA split, while centered variants enlarge prototype separation ($\Delta_P \approx 1.36$) with a trade-off in unfactual accuracy. Details of the compact table, algorithm, and LLM experiments are postponed to Appendix §A.2.1.

Both *semantic sufficiency* (decisions via leading subspaces) and *quantification* (one-dimensional energy) live entirely in Gram space. This yields an auditable, label-free, model-swap-stable scalar $Q$ for our CRC actuator in §4.

## 4 Batched Bootstrap CRC

### 4.1 Monotonicity–consistent actionable loss (policy–first design)

Our *Conformal Actuator* uses a single actionable loss paired with a calibration-only quality flag:

$$L(y, \lambda) \; = \; a_\lambda\big(Q(y)\big) \cdot m_\beta(y) \in [0, 1], \qquad R(\lambda) \; = \; \mathbb{E}[L(Y_{\mathrm{new}}, \lambda)] \,. \tag{4.1}$$

Here, $Q(y)$ is any scalar policy score; $a_\lambda : \mathbb{R} \to [0, 1]$ is a gate that is pointwise bounded and *monotone (non-increasing) in $\lambda$*; and $m_\beta(y) \in [0, 1]$ is an offline flag encoding the task's risk to be controlled.

The family $\{a_\lambda\}$ is the control mechanism—instantiated as a binary indicator, a quantile gate, or a smooth gate—under the sole assumption that it is monotone in $\lambda$. As $\lambda$ increases, the action moves consistently in one direction (e.g., becomes stricter). Consequently, $\lambda$ is the single tuning knob that carries ground-truth calibration into a physical actuator (escalate, re-route, regenerate, abstain), while requiring no labels at test time.

The flag $m_\beta$ is a bounded, task-chosen signal that marks outcomes to avoid when the policy acts (e.g., factuality errors upon acceptance, or over-unification that harms diversity). It is evaluated *only during calibration* using ground truth. CRC then learns the co-movement between this designated

---

[1]We implement the same rule in feature space using $C := V^\top H V$; spectral duality ensures equivalence to the item-space analysis, see Proposition A.3.

flag and the actionable policy by selecting $\hat{\lambda}$ to control $R(\lambda)$. At deployment, $m_\beta$ is not used; we apply the learned gate $a_{\hat{\lambda}}\big(Q(y)\big)$ in real time.

The gate $a_\lambda$ is label-free and can run on cheap signals (e.g., Gram-based measurements), while $m_\beta$ may be expensive/sparse/noisy and is used only in calibration. After tuning, no online labels are needed: we threshold a scalar policy score, which is compute-efficient, and with BB-CRC/RBWA, batching and bootstrap smoothing reduce LLM calls while preserving finite-sample validity.

## 4.2 BB-CRC AND RBWA-CRC

With $\ell_{\lambda,\beta}$ defined, our goal is to select a single global threshold $\hat{\lambda}$ that keeps the expected loss well bounded. Conformal Risk Control (CRC) provides such a finite-sample guarantee. However, when the loss depends on LLM outputs, naïve CRC can be costly because each assessment may require multiple model invocations. *Batched Bootstrap CRC (BB-CRC)* addresses both validity and efficiency by reusing a small held-out set and resampling it internally.

We split $n=GI$ instances into $G$ equal batches and, within each batch, draw $K$ bootstrap replicates from the same held-out data. A bias-corrected bootstrap average then yields a data-dependent threshold $\hat{\lambda}_Z$ that controls risk in finite samples. Practically, this delivers (i) **fewer LLM calls** at a fixed risk budget by recycling a batch via resampling, and (ii) **validity by design** maintaining exchangeability and theoretical guarantee.

**Lemma 4.1** (Distributional invariance). *Under the general assumptions,* $Y_{\text{new}} \mid \{Z_j^g : j = 1,\ldots,K;\ g = 1,\ldots,G\} \stackrel{D}{=} Y_{\text{new}} \sim \mathbb{P}_Y$ *(4.2), and, for each* $j = 1,\ldots,K$, $Z_j^{G+1} \mid \{Z_i^g : i=1,\ldots,K;\ g=1,\ldots,G\} \stackrel{D}{=} Z_j^{G+1} \sim \mathbb{P}_Y$ *(4.3).*

This Lemma 4.1 underpins the BB-CRC procedure. Given the calibration replicates, a new outcome and a "next-round" bootstrap replicate from an unused batch behave as draws from the same population. This lets us compare the new outcome to the bootstrapped world under exchangeability and motivates the BB-CRC desgin. We now present the BB-CRC algorithm.

---

**Algorithm 4.1** Batched Bootstrap Conformal Risk Control (BB-CRC)

---

1: **Input:** trajectories $\{Y_k\}_{k=1}^n$, batches $G$, replicates $K$, tolerance $\alpha$
2: Partition $\{Y_k\}_{k=1}^n$ into $\{B_g\}_{g=1}^G$ of equal size $I = n/G$
3: **for** $g = 1$ **to** $G$ **do**
4:     Draw $K$ bootstrap replicates $\{\mathbf{Z}_j^g\}_{j=1}^K$ from $B_g$
5: **end for**

6: $\hat{\lambda}_Z \leftarrow \inf\Big\{\lambda : \frac{1}{(G+1)K} \sum_{g=1}^{G} \sum_{j=1}^{K} L(\mathbf{Z}_j^g, \lambda) + \frac{1}{G+1} \leq \alpha \Big\} \wedge \lambda_{\max}$

7: **Return** $\hat{\lambda}_Z$

---

**Theorem 4.2** (Finite-sample BB-CRC). *Assume* $\{B_g\}_{g=1}^{G+1}$ *are i.i.d and* $\{Y_{g,1},\ldots,Y_{g,I}\}$ *are exchangeable for* $g = 1,2,\ldots,G+1$. *Let* $Y_{new} = Y_{n+1}$. *With loss L right-continuous w.r.t.* $\lambda$ *and bounded in* $[0,1]$ *and* $L(\cdot,\lambda_{max}) \leq \alpha$, *the estimator* $\hat{\lambda}_Z$ *returned by Algorithm 4.1 satisfies*

$$\mathbb{E}\big[L(Y_{\text{new}}, \hat{\lambda}_Z)\big] \leq \alpha.$$

Using Lemma 4.1, BB-CRC calibrates by contrasting held-out losses with the "next-batch" bootstrap world, yielding $\hat{\lambda}_Z$ with the guarantee in Theorem 4.2. We next generalize by replacing within-batch resampling with a single randomized convex combination across items. The *Randomized Batched Weighted Average CRC (RBWA-CRC)* method draws a simplex-valued weight vector $p_g$ per batch and computes a weighted mean of item losses in place of bootstrap replicates. This preserves finite-sample validity, introduces a transparent variance dial via the weight law, and enables mix-aware calibration—while leaving deployment unchanged (we still act via $a_{\hat{\lambda}}$).

---

**Algorithm 4.2** Randomized Batched Weighted Average Conformal Risk Control (RBWA-CRC)

1: **Input:** $\{Y_k\}_{k=1}^n$, batches $G$, weight law $\mathcal{P}_\mathcal{S}$, tolerance $\alpha$
2: Partition $\{Y_k\}$ into $\{B_g\}_{g=1}^G$ with $|B_g| = I = n/G$, $\{p_g\}_{g=1}^G$ are i.i.d.
3: **for** $g = 1{:}G$ **do**
4:      Sample $p_g = (p_{g,1}, \dots, p_{g,I}) \sim \mathcal{P}_\mathcal{S}$, independent of $B_g$
5:      Set $L_g(\lambda) = \sum_{i=1}^I p_{g,i} L(Y_{g,i}, \lambda)$
6: **end for**

7: $\hat{\lambda}_p \leftarrow \left( \inf\left\{ \lambda : \dfrac{1}{G+1} \sum_{g=1}^G L_g(\lambda) + \dfrac{1}{G+1} \le \alpha \right\} \right) \wedge \lambda_{\max}$

8: **Return** $\hat{\lambda}_p$

---

**Theorem 4.3** (Finite-sample RBWA-CRC)*. Assume $\{B_g\}_{g=1}^{G+1}$ are i.i.d and $\{Y_{g,1}, \dots, Y_{g,I}\}$ are exchangeable for $g = 1, 2, \dots, G+1$. Let $Y_{new} = Y_{n+1} = Y_{G+1,1}$. With loss $L$ right-continuous w.r.t. $\lambda$ and bounded in $[0, 1]$ and $L(\cdot, \lambda_{max}) \le \alpha$, the estimator $\hat{\lambda}_p$ returned by Algorithm 4.2 satisfies*

$$\mathbb{E}\big[L(Y_{\text{new}}, \hat{\lambda}_p)\big] \le \alpha.$$

**Remark: RBWA-CRC subsumes BB-CRC.** Let $\{w_j\}_{j=1}^K \overset{i.i.d.}{\sim} \text{Uniform}(\{1, \dots, I\})$ and set $u_i = \#\{j : w_j = i\}/K$, with $u = (u_1, \dots, u_I) \in \mathcal{S}$ and $\mathcal{P}_\mathcal{S}$ the law of $u$. Choosing $p_g \sim \mathcal{P}_\mathcal{S}$ in RBWA-CRC reproduces the BB-CRC resampling scheme within the RBWA template. Thus RBWA-CRC is a strict generalization: it retains the exchangeability logic, bias correction, and finite-sample validity, while replacing resampling with exogenous simplex weights that smooth and stabilize the empirical risk curve. In practice, design the loss once via equation 4.1, calibrate a single threshold with BB-CRC (bootstrap reuse) or RBWA-CRC (mix-aware weighted averaging), and deploy using the action rule alone.

### 4.3 WHY RANDOMIZED WEIGHTS HELP IN RBWA-CRC

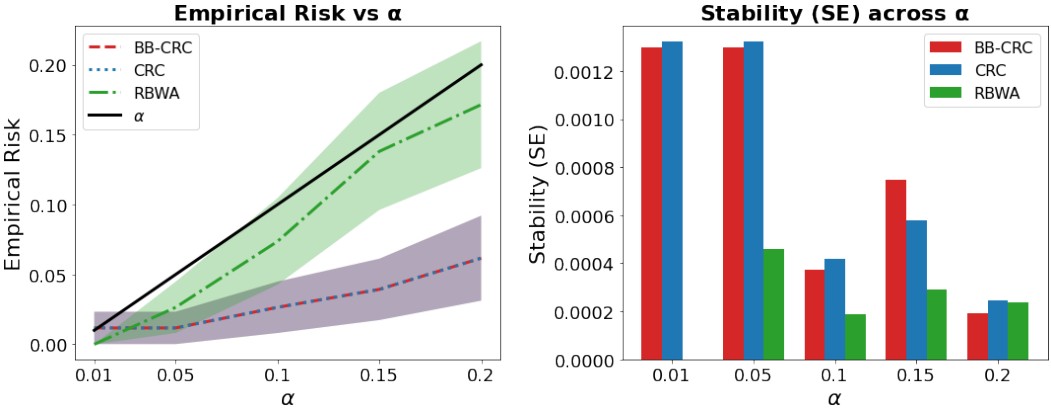

Figure 1: **Calibration comparison. Left:** Empirical risk at the calibrated threshold versus $\alpha$. RBWA closely tracks $y = \alpha$ as its property of unbiased smoothing and anti-concentration, while both BB-CRC and RBWA-CRC remain well-bounded and BBCRC is more conservative. **Right:** Stability (measured as standard error of $\hat{\lambda}$) versus $\alpha$. RBWA demonstrates superior threshold stability, while BBCRC attains a moderate enhancement in stability compared to standard CRC.

RBWA computes the per-batch statistic

$$L_g(\lambda) = \sum_{i=1}^I p_{g,i}\, \ell_{g,i}(\lambda), \qquad \ell_{g,i}(\lambda) = L(Y_{g,i}, \lambda) \in [0, 1], \quad p_g \in \mathcal{S},$$

with exogenous weights $p_g$ independent of $B_g$. The two theorems below show—without assuming a specific loss form—why this randomization stabilizes calibration: random weights act as an *unbiased smoother* with a single variance dial and remove lattice ties (anti-concentration). In practice, this smooths the CRC risk curve and yields more stable thresholds, without changing the actuator.

**Theorem 4.4** (RBWA moments: unbiased smoothing, variance dial, and anti-concentration). *Let $p_g \sim \mathrm{Dirichlet}(\eta \mathbf{1})$ with $\eta > 0$ and set $\kappa := I\eta$. For any fixed $\lambda$ and any bounded losses $\{\ell_{g,i}(\lambda)\}_{i=1}^I \subset [0,1]$:*

*(a)* Unbiasedness: $\mathbb{E}\big[L_g(\lambda) \mid \ell\big] = \mu(\lambda)$.

*(b)* Variance dial: $\mathrm{Var}\big(L_g(\lambda) \mid \ell\big) = \mathrm{Var}_{\mathrm{emp}}(\ell_g(\lambda))/(\kappa + 1)$. *Thus, for any $t > 0$,*

$$\Pr\left(|L_g - \mu| \geq t \mid \ell\right) \leq \frac{\mathrm{Var}_{\mathrm{emp}}(\ell_g)}{(\kappa + 1)t^2}, \qquad \Pr\left(L_g \geq \mu + t \mid \ell\right) \leq \frac{\mathrm{Var}_{\mathrm{emp}}(\ell_g)}{\mathrm{Var}_{\mathrm{emp}}(\ell_g) + (\kappa + 1)t^2}.$$

*(c)* Anti-concentration: *if $(\ell_1(\lambda), \ldots, \ell_I(\lambda))$ is not constant, then $L_g(\lambda)$ has no atoms $(\Pr(L_g = t \mid \ell) = 0$ for all $t$), hence threshold ties caused by discrete lattice values disappear.*

Keeping the weight precision $\kappa = I\eta$ roughly constant across folds makes dispersion comparable across iterations. A CLT then yields closed-form bands for $\bar{L}_G(\lambda)$ and supports a simple operational rule: choose the smallest $\lambda$ whose *upper* CLT band (plus the standard $+1/(G+1)$ correction) lies below $\alpha$.

**Theorem 4.5** (RBWA calibration CLT under precision stabilization). *Fix a $\lambda$. Assume batches are i.i.d., losses are bounded in $[0,1]$. Let $p_g \sim \mathrm{Dirichlet}(\mathbf{1})$. Let*

$$\mu(L) = \mathbb{E}[\mu_g], \; \mathrm{Var}(L) = \frac{\mathbb{E}[\mathrm{Var}_{\mathrm{emp}}(\ell_g)]}{+1} + \mathrm{Var}(\mu_g)$$

*they are well-defined because $\{\ell_g\}_{g=1}^G$ are i.i.d.. Assume $\mathrm{Var}(\ell)$ is finite. Then*

$$\sqrt{G}\left(\bar{L}_G - \mu(L)\right) \Rightarrow \mathcal{N}(0, \mathrm{Var}(L)), \qquad \bar{L}_G = \frac{1}{G}\sum_{g=1}^G L_g$$

*as $G \to \infty$.*

On LLM responses (ASQA) in Fig. 1(a), both BB-CRC and RBWA stay bounded by the risk budget, with RBWA aligning more closely to the target $y=\alpha$, while CRC and BB-CRC exhibit conservatism. This agrees with Theorems 4.4–4.5: Dirichlet randomization yields an *unbiased*, *anti-concentrated* batch loss, so $\bar{L}_G(\lambda)$ is smooth and the calibration constraint tends to be *active*, matching $\alpha$ up to CLT-scale fluctuations. In (Fig. 1 (b)), RBWA is observed to achieve the lowest standard error/optimal parameter stability of the calibrated threshold across $\alpha$.

# 5 EXPERIMENTS: LLM FACTUALITY LIFTING

## 5.1 DATA GENERATION PIPELINE AND METRICS

Two key questions are discussed: (i) Can our conformal actuator framework reduce LLM hallucination by improving factual accuracy across various datasets and contexts? (ii) Can the framework align an LLM-as-Judge score with factuality to make its randomness *measurable* in terms of factuality and reliability?

We evaluate across four complementary QA datasets, each surfacing a distinct failure mode: *ASQA*—ambiguity and under-specification (Stelmakh et al., 2023); *NQ-Open*—single-hop factoid retrieval (Lee et al., 2019; Kwiatkowski et al., 2019); *HotpotQA*—multi-hop composition (Yang et al., 2018); and *AmbigQA*—aliases and answer sets (Min et al., 2020). To probe sensitivity, we add two ablations: a decoding entropy stress test and a vendor swap.

For every open-domain QA query, we create a varied *response set* combining *plain* answers with structured *noise*, and assess each candidate using the clear metric **Factuality Severity** (FS). All

artifacts are kept provider-agnostic across OpenAI, Together, and Gemini (OpenAI et al., 2024; Grattafiori et al., 2024; Team et al., 2024). We hold the measurement pipeline fixed—decoding knobs, the counts of paraphrases and answers per item, and the normal-noise mix while spanning providers and model sizes (e.g., Llama-3.3-70B, Mixtral-8×7B, Llama-3.1-8B, GPT-4o-mini) (Grattafiori et al., 2024; Jiang et al., 2024; Grattafiori et al., 2024; OpenAI et al., 2024). Separating *what we measure* from *what we vary* (datasets, temperatures, and models) shows that conclusions do not hinge on any single setting: The criteria for being "far from truth" and "out of consensus" are consistently maintained across various tasks and providers.

To measure deviation from references, we employ a BERTScore-F1 (Zhang et al., 2020) adjusted to the baseline focusing on *answer head*. Define $R_q$ as the paraphrased reference set for a given question $q$, and $\mathrm{head}(a)$ as the candidate's head. We introduce **Factuality Severity (FS)** as

$$\mathrm{FS}(a) \;=\; 1 - \max_{r \in R_q} \mathrm{BERTScoreF1}\big(\mathrm{head}(a),\, r\big) \in [0, 1]. \tag{5.1}$$

$\mathrm{FS}(a) = 0$ signals exact alignment with the reference, indicating the response is essentially a paraphrase. Scores near 1 imply semantic divergence. Prioritizing response head reduces bias from reasoning and length.

An LLM judge gives a rubric-based score $J(a) \in [0, 100]$ to the answer head (correctness, faithfulness, completeness, clarity; G-Eval style) (Liu et al., 2023a); we normalize this as $J_{\mathrm{norm}}(a) = J(a)/100$ and define **LLM-as-Judge Severity (JS)** as $\mathrm{JS}(a) = 1 - J_{\mathrm{norm}}(a)$ ranging from 0 to 1.

## 5.2 Factuality Lifting on Actionable Policy

We retain the same policy–first loss:

$$\mathcal{L}(y, \lambda) = a_\lambda\big(Q(y)\big) \cdot m(y), \qquad a_\lambda(u) = \mathbf{1}\{u \geq \lambda\}, \tag{5.2}$$

where $m(y) \in [0, 1]$ is an *offline* factuality severity used only for calibration and $Q(y)$ is a *label-free, online* policy score. At deployment we compute $Q(y)$, apply the gate $a_{\hat{\lambda}}(Q(y))$, and never read $m(y)$. The single knob $\lambda$ therefore maps statistical calibration into a physical action (ship/abstain/regen/escalate), while cleanly separating *measurement* ($m$) from *action* ($Q$). We instantiate two choices for the online score:

- **(P1) Gram–energy consensus** $Q_E(y) = E(y) \in [0, 1]$: a row-energy signal from the response-queue centered Gram geometry (cf. Eq. 2.4), cheap and label-free.
- **(P2) LLM-as-Judge** $Q_J(y) = J_{\mathrm{norm}}(y) \in [0, 1]$: a rubric grade on the answer head from a light grader (G-Eval style).

Across four QA datasets (ASQA, NQ-Open, HotpotQA, AmbigQA) and two ablations, we hold the measurement pipeline fixed and vary dataset, temperature, and provider. Under these controlled variations, both *policy* scores, Gram energy $Q_E$ and LLM-as-Judge $Q_J$, lift factuality, with $Q_E$ exhibiting more *uniform* improvements across tasks and risk budgets (Fig. 2). By design, $Q_E$ is a consensus-seeking, outputs-only signal derived from centered Gram geometry: it is normalized to $[0, 1]$, permutation-invariant, and—in entropy stress tests—suppresses isolated outliers while concentrating acceptance on dense semantic modes; in vendor/model swaps, its dependence on outputs plus a fixed encoder preserves acceptance regions and supports a portable, API-level control layer. $Q_J$ remains operationally useful wherever a rubric is available; paired with CRC, its score becomes measurable and risk-trackable, though its FS lift attenuates at larger $\alpha$.

## 5.3 Baseline

| Name | Policy score $Q$ | Gate / Calibration |
|------|------------------|---------------------|
| G-Eval-Naive | Judge score $Q_N = J_{\mathrm{norm}} \in [0, 1]$ | Fixed $\lambda \in \{0.99, 0.95, 0.90, 0.85, 0.80\}$ (no guarantees) |
| G-Eval-CRC | Judge score $Q_J = J_{\mathrm{norm}}$ | BB-CRC threshold $\hat{\lambda}(\alpha)$ (finite-sample validity) |
| Gram-CRC | Gram energy $Q_E = E \in [0, 1]$ | RBWA-CRC threshold $\hat{\lambda}(\alpha)$ (finite-sample validity) |

Table 1: **Mode summary.** Modes vary in online score $Q$ and calibration (fixed vs. CRC ).

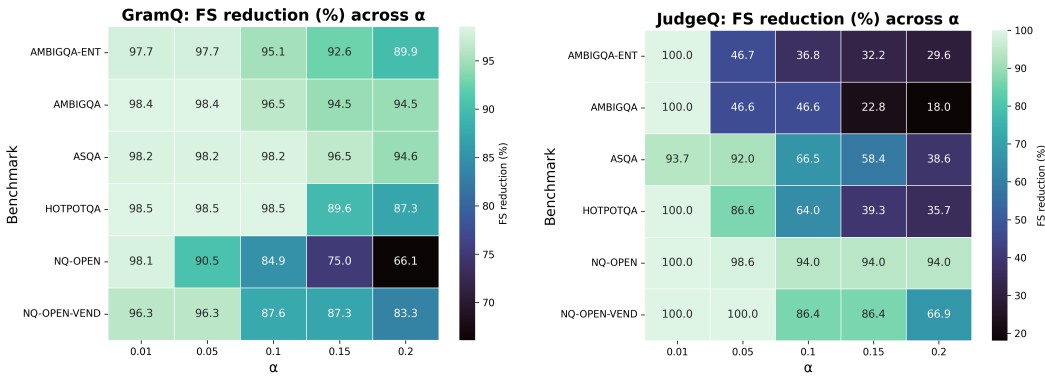

(a) Policy $Q_E$: FS reduction (%) per benchmark.      (b) Policy $Q_J$: FS reduction (%) per benchmark.

Figure 2: **Factuality lifting across diversified settings.** Heatmaps show the % drop in $FS$ from *Unshipped* to *Shipped* under the same gate $a_{\hat{\lambda}}$; rows are benchmarks (incl. ablations), columns are risk budgets $\alpha$; left/right panels differ only by the policy score $Q$ ($Q_E$ vs. $Q_J$). With $Q_E$ (left), reductions are high and notably *uniform* across datasets and $\alpha$, remaining stable under entropy stress and provider/model swaps. With $Q_J$ (right), the judge-based policy also yields substantial gains, with lift varying more by task and budget.

To baseline the actuator, we ablate along two axes: (i) the online *policy score* $Q \in \{Q_J, Q_E\}$ and (ii) how the threshold $\lambda$ is set (fixed versus CRC). This yields three deployment modes that all use the *same* one-knob gate $a_\lambda$ but differ only in the score and calibration (Table 1).

| Method | $\alpha=0.01$ | 0.05 | 0.10 | 0.15 | 0.20 |
|---|---|---|---|---|---|
| G-Eval-Naive | 12.8 | 9.7 | 8.9 | 9.1 | 9.0 |
| G-Eval-CRC | 98.9 | 78.3 | 65.3 | 55.0 | 46.5 |
| Gram-CRC | 97.9 | 96.7 | 93.6 | 89.4 | 86.0 |

Table 2: **FS reduction (%) across risk budgets** $\alpha$. Entries are the percentage drop in $FS$; higher is better. Moving from a fixed judge threshold to CRC (*G-Eval-Naive→G-Eval-CRC*) shows the gain from calibration, while switching the policy score to Gram energy (*G-Eval-CRC→Gram-CRC*) yields the strongest and most uniform lift—for example, at $\alpha=0.20$ the reductions are $86.0\%$ (Gram-CRC) vs. $46.5\%$ (G-Eval-CRC) vs. $\approx 9\%$ (G-Eval-Naive).

Crossing *policy* ($Q_E$ vs. $Q_J$) with *calibration* (fixed threshold vs. CRC) yields three modes that *share the same actuator* $a_\lambda$: *G-Eval-Naive* (pure baseline; fixed judge threshold, no guarantees), *G-Eval-CRC* (judge policy made risk-controlled), and *Gram-CRC* (full geometry policy). This design serves two purposes. *First*, **G-Eval-CRC** is both a strong baseline against **Gram-CRC** *and* our instrument for controlling LLM-as-judge randomness: calibration turns the judge score into a *measurable, risk-tracking* knob. *Second*, **G-Eval-Naive** isolates the value of calibration itself. Table 2 quantifies the two-step story: *Naive→CRC* captures the gain from *validity* (e.g., $9\% \rightarrow 46.5\%$ FS reduction at $\alpha=0.20$), while *G-Eval-CRC→Gram-CRC* captures the gain from the *policy signal* (to $86.0\%$ at $\alpha=0.20$). Our conformal risk control framework maintains risk within budget and stabilizes thresholds, allowing for *single calibration and frequent deployment with controlled LLM randomness*.

## 6 CONCLUSION

We introduce a concise *calibrate-once, deploy-often* framework for controlling risk in black-box LLMs. This model operates using a single scalar *actuator* with a unified monotone threshold. The Conformal Risk Control (CRC) methodology provides finite-sample assurances within a specified risk level $\alpha$. Two variants further strengthen reliability and efficiency: **BB-CRC** (batched bootstrap CRC) boosts data efficiency by pooling across bootstrap splits, and **RBWA-CRC** (randomized

batch weight) minimizes threshold variance, enhancing deployment stability. Alongside CRC, our *Gram geometry sufficiency* principle swiftly provides auditable uncertainty quantification at the API boundary by converting complex semantics into dependable metrics. Taken together, these pieces constitute a *general* and *portable* template for risk control: any task with a monotone loss can inherit the same actuator-and-threshold mechanism, making our approach immediately extensible beyond LLM setting to broader risk control problems.

In real-world LLM settings, the calibrated actuator systematically tames *stochastic generative variability* in black-box models. The actuator meets target risk budgets and produces consistent *factuality lift*, thereby mitigating hallucination without token-level probabilities or labels. Beyond generation, the same actuator enables *LLM-as-judge* routing and triage: it makes judge pipelines measurable, portable across models, and auditable for production governance. In short, a single calibrated actuator turns LLM variability into validity: geometry ($Q_E$) provides provider-agnostic gains, and CRC makes those gains allocatable at a user-chosen risk budget.

**Limitations & Future Work.** We highlight two directions. **(1) Relaxed exchangeability.** Our guarantees rest on exchangeability; relaxing this assumption to handle covariate shift, prompt drift, and temporal dependence is a key next step. **(2) LLM-as-judge at scale.** We aim to broaden the judge setting from QA factuality to pairwise ranking, critique grading, safety adjudication, and multi-judge ensembling, exploring how $Q_E$ and CRC interact with rubric design, aggregation, and adversarial prompting.

**Use of AI for language editing.** We used OpenAI ChatGPT and Overleaf Writefull solely for language polishing (grammar, clarity, and style) of author-written text. All ideas, experiments, and conclusions are the authors' own. The authors reviewed and verified all content and take full responsibility for any errors.

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

# A    PROOFS

## A.1    SECTION 3: PROOFS, SELF-CONSISTENCY LINK, DUALITY

Unit-norm embeddings $v_i = \psi(y_i) \in \mathbb{R}^d$; $V \in \mathbb{R}^{n \times d}$ with rows $v_i^T$; $G = VV^\top$; $H = I_n - \frac{1}{n}\mathbf{1}\mathbf{1}^\top$; $\tilde{G} = HGH$.

**Setup and notation.**    Let $\tilde{G} \in \mathbb{R}^{n \times n}$ be the centered sample Gram matrix computed from a test batch of $n$ vectors. For each regime (class) $k \in \{1, \ldots, K\}$, take $M_k$ calibration batches of size $n$ and compute their centered Grams $\tilde{G}_k^{(m)}$ for $m = 1, 2, ..., M_k$. Denote $S_k := \mathbb{E}[\tilde{G} \mid k]$, decomposed as $S_k = U_k \Lambda_k U_k^\top$, which is estimated by

$$\hat{S}_k = \frac{1}{M_k} \sum_{m=1}^{M_k} \tilde{G}_k^{(m)}, \qquad \hat{S}_k = \hat{U}_k \hat{\Lambda}_k \hat{U}_k^\top$$

and assume an eigengap at rank $r$:

$$\gamma_k = \lambda_r(S_k) - \lambda_{r+1}(S_k) > 0.$$

Let $P_k := U_k^{(r)}(U_k^{(r)})^\top$ be the (theoretical) rank-$r$ population projector for class $k$. From calibration data, form empirical prototype projectors $\hat{P}_k$ (at rank $r$): for the calibration batch let $\hat{U}_k^{(r)}$ be the top-$r$ eigenvectors (corresponding to the $r$ top eigenvalues) of $\hat{S}_k$, and define $\hat{P}_k = \hat{U}_k^{(r)}(\hat{U}_k^{(r)})^\top$. Similarly, for the test batch let let $\hat{U}^{(r)}$ be the top-$r$ eigenvectors of $\tilde{G}$ and define $\hat{P} = \hat{U}^{(r)}(\hat{U}^{(r)})^\top$ the sample projector. Define the between-class separation (on prototypes)

$$\Delta_P = \min_{j \neq \ell} \|\hat{P}_j - \hat{P}_\ell\|_F.$$

**Classifier**

$$\hat{k} = \arg\max_k \langle \hat{P}, \hat{P}_k \rangle_F. \tag{A.1}$$

This depends on the test data only through $\tilde{G}$ (via $\hat{P}$) and the stored Gram-space prototypes $\{\hat{P}_k\}$.

### A.1.1    PROJECTOR PERTURBATION AND SEMANTIC SUFFICIENCY (MAIN-TEXT THEOREM 3.1)

**Lemma A.1** (Davis–Kahan projector perturbation; Frobenius form). *If $\|\tilde{G} - S_k\|_{\mathrm{op}} \leq \varepsilon$, then the top-$r$ projector $\hat{P}$ of $\tilde{G}$ obeys $\|\hat{P} - P_k\|_F \leq \frac{2\sqrt{r}}{\gamma_k}\varepsilon$.*

*Proof.* Let $\Theta = \mathrm{diag}(\theta_1, \ldots, \theta_r)$ be the diagonal matrix of principal angles between the subspaces $\mathsf{span}(\hat{U}^{(r)})$ and $\mathsf{span}(U_k^{(r)})$. The Davis–Kahan $\sin\Theta$ theorem (Yu et al., 2015) gives the operator-norm bound

$$\|\sin\Theta\|_{\mathrm{op}} \leq \frac{\|\tilde{G} - S_k\|_{\mathrm{op}}}{\gamma_k} \leq \frac{\varepsilon}{\gamma_k}.$$

Hence, by $\|\cdot\|_F \leq \sqrt{r}\|\cdot\|_{\mathrm{op}}$,

$$\|\sin\Theta\|_F \leq \sqrt{r}\|\sin\Theta\|_{\mathrm{op}} \leq \frac{\sqrt{r}}{\gamma_k}\varepsilon.$$

For rank-$r$ orthogonal projectors $P_k$ and $\hat{P}$ associated with $U_k^{(r)}$ and $\hat{U}^{(r)}$,

$$\|\hat{P} - P_k\|_F^2 = \mathrm{tr}((\hat{P} - P_k)^\top(\hat{P} - P_k)) = \mathrm{tr}(\hat{P}) + \mathrm{tr}(P_k) - 2\,\mathrm{tr}(\hat{P}P_k)$$
$$= 2r - 2\,\mathrm{tr}((\hat{U}^{(r)})^\top U_k^{(r)}(U_k^{(r)})^\top(\hat{U}^{(r)}))$$

The singular values of $\hat{U}^{(r)})^\top U_k^{(r)}$ are $\cos\theta_1, \ldots, \cos\theta_r$, the cosines of the principal angles, hence

$$\mathrm{tr}((\hat{U}^{(r)})^\top U_k^{(r)}(U_k^{(r)})^\top(\hat{U}^{(r)})) = \|(\hat{U}^{(r)})^\top U_k^{(r)}\|_F^2 = \sum_{i=1}^r \cos^2\theta_i.$$

Therefore

$$\|\widehat{P} - P_k\|_F^2 = 2\sum_{i=1}^{r}(1 - \cos^2\theta_i) = 2\sum_{i=1}^{r}\sin^2\theta_i = 2\|\sin\Theta\|_F^2,$$

then

$$\|\widehat{P} - P_k\|_F = \sqrt{2}\|\sin\Theta\|_F \leq \sqrt{2}\cdot\frac{\sqrt{r}}{\gamma_k}\varepsilon_n < \frac{2\sqrt{r}}{\gamma_k}\varepsilon$$

$\square$

**Theorem A.2** (Semantic sufficiency of Gram projectors). *Under the conditions stated in Theorem 3.1 of the main text, the spectral-overlap rule selects the correct class with margin $m \geq \Delta_P^2/4 > 0$.*

*Proof.* By Lemma and the concentration assumption $\|\tilde{G} - S_{k^\star}\|_{\mathrm{op}} \leq \varepsilon_n$,

$$\|\hat{P} - P_{k^\star}\|_F \leq \frac{2\sqrt{r}}{\gamma_{k^\star}}\varepsilon_n.$$

Triangle inequality then gives

$$\|\hat{P} - \hat{P}_{k^\star}\|_F \leq \|\hat{P} - P_{k^\star}\|_F + \|P_{k^\star} - \hat{P}_{k^\star}\|_F \leq \frac{2\sqrt{r}}{\gamma_{k^\star}}\varepsilon_n + \delta_{\mathrm{proto}} =: \rho.$$

Recall for rank-$r$ orthogonal projectors $A, B$ we have $\|A\|_F^2 = \|B\|_F^2 = r$ and

$$\langle A, B\rangle_F = r - \frac{1}{2}\|A - B\|_F^2.$$

Thus, for any $j \neq k^\star$,

$$\langle\hat{P}, \hat{P}_{k^\star}\rangle_F - \langle\hat{P}, \hat{P}_j\rangle_F = \frac{1}{2}\big(\|\hat{P} - \hat{P}_j\|_F^2 - \|\hat{P} - \hat{P}_{k^\star}\|_F^2\big).$$

By the reverse triangle inequality $\|\hat{P} - \hat{P}_j\|_F \geq \|\hat{P}_j - \hat{P}_{k^\star}\|_F - \|\hat{P} - \hat{P}_{k^\star}\|_F$. Let $\Delta_P = \min_{j\neq k^\star}\|\hat{P}_j - \hat{P}_{k^\star}\|_F$. Then for every $j \neq k^\star$,

$$\langle\hat{P}, \hat{P}_{k^\star}\rangle_F - \langle\hat{P}, \hat{P}_j\rangle_F \geq \frac{1}{2}\big(\Delta_P - \rho\big)^2 - \frac{1}{2}\rho^2 = \frac{1}{2}\big(\Delta_P^2 - 2\Delta_P\rho\big) = \frac{\Delta_P}{2}(\Delta_P - 2\rho).$$

Consequently, if $\rho < \frac{1}{2}\Delta_P$, which is assured by condition equation 3.4, then every right-hand side is positive and hence $\langle\hat{P}, \hat{P}_{k^\star}\rangle_F > \langle\hat{P}, \hat{P}_j\rangle_F$ for all $j \neq k^\star$, so $\hat{k} = k^\star$. Finally, if the stronger condition equation 3.4 holds then $\rho \leq \frac{1}{4}\Delta_P$, so $\Delta_P - 2\rho \geq \frac{1}{2}\Delta_P$ and therefore the overlap margin satisfies

$$m = \min_{j\neq k^\star}\big\{\langle\hat{P}, \hat{P}_{k^\star}\rangle_F - \langle\hat{P}, \hat{P}_j\rangle_F\big\} \geq \frac{\Delta_P}{2}\cdot\frac{\Delta_P}{2} = \frac{\Delta_P^2}{4}.$$

This completes the proof. $\square$

### A.1.2 SPECTRAL DUALITY (ITEM VS. FEATURE SPACE)

**Proposition A.3** (Spectral duality). *Let $V \in \mathbb{R}^{n\times d}$, $H := I_n - \frac{1}{n}\mathbf{1}\mathbf{1}^\top$, and $Z := HV$. Let $Z = U\Sigma W^\top$ be a compact SVD with $U \in \mathbb{R}^{n\times r}$, $\Sigma \in \mathbb{R}^{r\times r}$, $W \in \mathbb{R}^{d\times r}$. Define $S := ZZ^\top$, $C := Z^\top Z$, and projectors*

$$P_S := U_r U_r^\top, \qquad P_C := W_r W_r^\top.$$

*Then $S$ and $C$ share the same nonzero singular spectrum, and*

$$\langle P_S^{(a)}, P_S^{(b)}\rangle_{\mathrm{F}} = \big\|U_r^{(a)\top}U_r^{(b)}\big\|_{\mathrm{F}}^2, \qquad \langle P_C^{(a)}, P_C^{(b)}\rangle_{\mathrm{F}} = \big\|W_r^{(a)\top}W_r^{(b)}\big\|_{\mathrm{F}}^2.$$

*Here $(a)$ and $(b)$ index two different batches of centered data, each with its own SVD and associated projector.*

*Proof.* Since $ZW_r = U_r\Sigma_r$ and $Z^\top U_r = W_r\Sigma_r$, it follows that $U_r = ZW_r\Sigma_r^{-1}$ and $W_r = Z^\top U_r\Sigma_r^{-1}$, which yields the claimed identities. The overlap formulas follow from $\langle A, B\rangle_{\mathrm{F}} = \mathrm{tr}(A^\top B)$ and standard properties of principal angles. $\square$

**Corollary A.4** (Feature-space sufficiency). *Let $\widetilde{C} := Z^\top Z$ and $C_k := \mathbb{E}[\widetilde{C} \mid k]$ with eigengap $\gamma_k^{(C)} = \lambda_r(C_k) - \lambda_{r+1}(C_k) > 0$. Let $Q_k$ be the rank-$r$ projector of $C_k$, $\widehat{Q}_k$ the prototype projector from calibration, and $\widehat{Q}$ the test projector from $\widetilde{C}$. Define $\Delta_Q = \min_{j\neq\ell}\|\widehat{Q}_j - \widehat{Q}_\ell\|_{\mathrm{F}}$. If*

$$\|\widetilde{C} - C_{k^\star}\|_{\mathrm{op}} \leq \varepsilon_n, \qquad \frac{2\sqrt{r}}{\gamma_{k^\star}^{(C)}}\varepsilon_n + \|\widehat{Q}_{k^\star} - Q_{k^\star}\|_{\mathrm{F}} < \tfrac{1}{4}\Delta_Q,$$

*then*

$$\widehat{k} = \arg\max_k \langle\widehat{Q}, \widehat{Q}_k\rangle_{\mathrm{F}}$$

*recovers the correct class $\widehat{k} = k^\star$ with overlap margin at least $\Delta_Q^2/4$.*

*Proof.* Same as the proof of Theorem 2.1, with $ZZ^\top$ replaced by $Z^\top Z$ and left singular subspaces replaced by right singular subspaces. $\square$

### A.1.3 INTERACTION-ENERGY RANGE (QUANTIFICATION)

**Theorem A.5** (Unit-norm interaction–energy bound). *If $\|v_i\|_2 = 1$ for all $i$, then $1 \leq e(i; G) \leq \sqrt{n}$, with $e = \sqrt{n}$ when all $v_j$ align with $v_i$ and $e = 1$ when $v_i \perp v_{j\neq i}$.*

*Proof.* With $\|v_j\|_2 = 1$, the $j = i$ term in equation 3.1 gives $f(i; G)^2 \geq 1$. Since $|\langle v_i, v_j\rangle| \leq 1$,

$$f(i; G)^2 = \sum_{j=1}^{n}\langle v_i, v_j\rangle^2 \leq n,$$

hence $f(i; G) \leq \sqrt{n}$. Both bounds are attainable by orthogonality (lower) and equality (upper) configurations. $\square$

## A.2 SECTION 3: ALGORITHM AND RESULTS

### A.2.1 GRAM–PROJECTOR SPECTRAL-OVERLAP (GPSO) CLASSIFIER — INTUITION & IMPLEMENTATION

Given a small batch of responses, we embed each answer head (unit-norm) and form a Gram geometry that is (i) permutation-invariant over items, (ii) label-free at test time, and (iii) stable under model swaps. The decision lives in the leading *Gram subspace*: we compare the test batch's top-$r$ projector to calibrated prototype projectors via *spectral overlap*. Under a concentration assumption and prototype separation, the overlap rule recovers the correct class with a positive margin (Theorem 3.1 main text).[2]

**Notation.** Unit-norm embeddings $v_i = \psi(y_i) \in \mathbb{R}^d$, batch matrix $V \in \mathbb{R}^{n\times d}$, centering

$$H = I_n - \tfrac{1}{n}\mathbf{1}\mathbf{1}^\top, Z := HV,$$

item-space Gram $\tilde{G} = ZZ^\top = H(VV^\top)H$, feature-space scatter $\widetilde{C} = Z^\top Z$.

For class $k$, the projector is $P_k$ (item-space) or $Q_k$ (feature-space), with prototypes $\hat{P}$ or $\hat{Q}$; a test batch yields $\hat{P}$ or $\hat{Q}$. The spectral-overlap rule chooses

$$\hat{k} = \arg\max_k\langle\hat{P}, \hat{P}_k\rangle_F \quad \text{or equivalently with feature-space projectors}$$

---

[2]Feature/item-space duality ensures the same procedure works in feature space with $C = Z^\top Z$ (Proposition A.3).

**Centering and $L^2$.** L2 normalization removes scale/length bias in encodings; centering ($H$) removes the rank-one mean component so that leading directions represent *consensus deviations* rather than the global mean. In practice, centering enlarges prototype separation $\Delta_P$ but can also change the best operating rank $r$ and the unfactual class geometry; we therefore report both centered and non-centered pipelines (cf. ablations A/B/C/D below).

We implement in *feature space* by default (numerically cheaper when $d \ll n$):

$$\widetilde{C} = \begin{cases} V^\top H V & \text{(centered)} \\ V^\top V & \text{(no centering)} \end{cases}, \qquad \hat{Q} = \mathrm{proj}_r(\widetilde{C}), \qquad s_k = \langle \hat{Q}, \bar{Q}_k \rangle_F.$$

*Rank $r$ selection by eigengap.* On2 class-average scatter (or bootstrap average) we pick $r = \arg\max_j(\lambda_j - \lambda_{j+1})$ subject to $1 \le r \le r_{\max}$, then re-project any averaged projector back to rank $r$.

---

**Algorithm A.1** GPSO (Calibration + Inference; feature-space implementation)

---

1: **Inputs:** Embeddings $V \in \mathbb{R}^{n \times d}$ (rows unit-norm), class label $\in \{\mathsf{good}, \mathsf{bad}\}$, prototypes $\{\bar{Q}_k\}$, rank cap $r_{\max}$.
2: **Preprocess:** Optionally center with $H = I - \frac{1}{n}\mathbf{1}\mathbf{1}^\top$; set $\widetilde{C} \leftarrow V^\top H V$ (or $V^\top V$ if no centering).
3: **Rank choice:** On calibration, average class scatters to $C_k^{\mathrm{bar}}$ (or bootstrap-average); choose $r_k$ by eigengap; set final $r \leftarrow \min_k r_k$.
4: **Prototype(s):** For each class $k$, collect projectors $Q_k^{(b)}$ from calibration or bootstrap replicates, average $\bar{Q}_k^{\mathrm{raw}} \leftarrow \frac{1}{B}\sum_b Q_k^{(b)}$, then project back to rank $r$: $\bar{Q}_k \leftarrow \mathrm{proj}_r(\bar{Q}_k^{\mathrm{raw}})$.
5: **Test projector:** $\hat{Q} \leftarrow \mathrm{proj}_r(\widetilde{C})$.
6: **Scores & decision:** $s_k \leftarrow \langle \hat{Q}, \bar{Q}_k \rangle_F$, $\hat{k} \leftarrow \arg\max_k s_k$, $m \leftarrow s_{\hat{k}} - \max_{j \neq \hat{k}} s_j$.
7: **Diagnostics (logged):** prototype separation $\Delta_P = \|\bar{Q}_{\mathsf{good}} - \bar{Q}_{\mathsf{bad}}\|_F$; concentration $\varepsilon = \|C_{\mathrm{test}} - C_k^{\mathrm{bar}}\|_{\mathrm{op}}$; eigengap $\gamma_k$ at $r$; prototype dispersion $\delta_{\mathrm{proto}} = \mathrm{median}_b\|Q_k^{(b)} - \bar{Q}_k\|_F$; margin-condition flag $[(2\sqrt{r}/\gamma_k)\varepsilon + \delta_{\mathrm{proto}} < \Delta_P/4]$; normalized margin $m/r$.

---

**Within-question (stratified).** For each question $q$, split $\mathsf{good}/\mathsf{bad}$ into train/test (ratio $0.6/0.4$ by default), build prototypes on train with bootstrap replicates ($B = 8$), and test on the held-out items. Repeat $n_{\mathrm{splits}} = 5$ times per $q$ with a fixed seed; report per-split and per-question means.

**Pipelines (ablations).** We compare three scatter constructions that isolate the role of centering and $L^2$:

(A) **L2+centered** (*Good Gram*): $V$ row-normalized, $\widetilde{C} = V^\top H V$.

(B) **L2/no-center**: $V$ row-normalized, $\widetilde{C} = V^\top V$.

(C) **no-L2+center**: raw rows, $\widetilde{C} = V^\top H V$.

(D) **no-L2+no-center**: raw rows, $\widetilde{C} = V^\top V$.

*Remark (duality).* Item-space GPSO with $\tilde{G}$ and feature-space GPSO with $\widetilde{C}$ are spectrally equivalent; our implementation adopts $\widetilde{C}$ for efficiency while the theory in §3 and App. A.1 is stated in $\tilde{G}$ for clarity (Proposition A.3).

### A.2.2 EXPERIMENT SETTINGS AND COMPACT RESULTS (FACTUAL VS. UNFACTUAL QA; GROUPED CV)

Each row is an answer head with fields `question`, `text`, and `forced_generation` (Boolean). We treat $\mathsf{good} \equiv$ (`forced_generation=False`) and $\mathsf{bad} \equiv$ (`True`). Embeddings use `all-MiniLM-L6-v2` with L2 row normalization unless disabled by the pipeline. All runs are seeded and grouped by question to prevent leakage.

For each fold we record: predicted class for held-out good/bad batches; unnormalized and normalized margins ($m$, $m/r$); rank $r$; prototype separation $\Delta_P$; concentration $\varepsilon$ (spectral norm $\|\widetilde{C}_{\text{test}} - C_k^{\text{bar}}\|_{\text{op}}$); eigengap $\gamma_k$; prototype dispersion $\delta_{\text{proto}}$; and a Boolean *margin-condition pass* flag

$$\underbrace{\frac{2\sqrt{r}}{\gamma_k}\varepsilon + \delta_{\text{proto}}}_{\text{LHS}} < \underbrace{\tfrac{1}{4}\Delta_P}_{\text{RHS}}.$$

Global summaries report macro and per-class accuracies, mean normalized margins, mean $\Delta_P$, average rank, and the fraction of folds that satisfy the margin condition.

*Within-question* uses $60\%/40\%$ stratified train/test with $n_{\text{splits}} = 5$ and bootstrap $B = 8$ for prototype stability; *LOO* uses other questions as calibration pools (skipping low-count classes) and tests on the held-out question.

Table 3: GPSO across datasets: macro/class accuracies and prototype distance $\Delta_P$.

| Dataset | Pipeline | Macro acc | Acc (factual) | Acc (unfactual) | $\Delta_P$ |
|---|---|---|---|---|---|
| ASQA | (A) L2+centered | 0.856 | 0.956 | 0.756 | 1.427 |
| ASQA | (B) L2/no-center | **0.972** | 1.000 | 0.944 | 1.412 |
| ASQA | (C) no-L2+center | 0.922 | 0.956 | 0.889 | 1.442 |
| ASQA | (D) no-L2/no-center | 0.967 | 1.000 | 0.933 | 1.412 |
| HotpotQA | (A) L2+centered | 0.924 | 0.939 | 0.909 | 1.430 |
| HotpotQA | (B) L2/no-center | **0.977** | 1.000 | 0.955 | 1.160 |
| HotpotQA | (C) no-L2+center | 0.955 | 0.955 | 0.955 | 1.415 |
| HotpotQA | (D) no-L2/no-center | 0.955 | 1.000 | 0.909 | 1.161 |
| NQ-Open | (A) L2+centered | 0.947 | 1.000 | 0.894 | 1.463 |
| NQ-Open | (B) L2/no-center | **0.970** | 1.000 | 0.939 | 0.932 |
| NQ-Open | (C) no-L2+center | 0.955 | 0.939 | 0.970 | 1.451 |
| NQ-Open | (D) no-L2/no-center | **0.970** | 0.985 | 0.955 | 0.924 |

**Findings.** (i) **Best macro accuracy** is consistently achieved by L2/no-center (B), which preserves length-free directions while letting the mean component contribute discriminative variance in this binary factuality task. (ii) **Centering increases prototype separation** ($\Delta_P \approx 1.36$ for A/C) but trades off with unfactual accuracy (bad-class geometry differs once the mean is removed). (iii) The **margin condition is auditable**: folds with larger $\Delta_P$ and stable prototypes (small $\delta_{\text{proto}}$) show higher normalized margins and a higher fraction of passes.[3]

**Interpretation.** Pipelines with centering and L2 (A/C) align with the spectral theory: they enlarge prototype separation $\Delta_P$ and yield cleaner subspace structures by removing length and mean effects. Yet, in practice, preserving the mean (B/D) consistently improves the accuracy, suggesting that the mean embedding itself carries label-related signals. This tension indicates that centering may enhance interpretability and theoretical guarantee, while non-centering may better capture dataset-specific features.

Compact algorithm and summary appear in Appendix A.1.4–A.1.5 of the paper; §3 presents the sufficiency theorem and its margin bound, and §4 connects the Gram geometry to a 1-D consensus score used by CRC.

---

[3]These diagnostics mirror Theorem 3.1's sufficient condition and are logged by the evaluator for each fold.

### A.3 PROOFS FOR SECTION 4

**Lemma 4.1** (Distributional invariance). *Under the general assumptions, $Y_{\text{new}} \mid \{Z_j^g : j = 1, \ldots, K; \; g = 1, \ldots, G\} \overset{D}{=} Y_{\text{new}} \sim \mathbb{P}_Y$ (4.2), and, for each $j = 1, \ldots, K$, $Z_j^{G+1} \mid \{Z_i^g : i = 1, \ldots, K; \; g = 1, \ldots, G\} \overset{D}{=} Z_j^{G+1} \sim \mathbb{P}_Y$ (4.3).*

*Proof.* Equation equation 4.2 follows immediately because $Y_{\text{new}}$ is independent of $\{Y^g\}_{g=1}^G$. For equation 4.3 we need only verify that the marginal law of $Z_j^{G+1}$ equals $\mathbb{P}_Y$. For any $y \in \mathbb{R}$,

$$\mathbb{P}(Z_j^{G+1} \leq y) = \mathbb{E}\big[\mathbf{1}\{Z_j^{G+1} \leq y\}\big]$$
$$= \sum_{k=1}^K g_k \, \mathbb{P}\big(Y_k^{G+1} \leq y\big)$$
$$= \mathbb{P}\big(Y \leq y\big),$$

here $g_k$ stand for the probability $\mathbb{P}(Z_j^{G+1} = Y_k^{G+1})$. Thus $Z_j^{G+1} \sim \mathbb{P}_Y$, completing the proof. $\square$

**Theorem 4.2** (Finite-sample BB-CRC). *Assume $\{B_g\}_{g=1}^{G+1}$ are i.i.d and $\{Y_{g,1}, \ldots, Y_{g,I}\}$ are exchangeable for $g = 1, 2, \ldots, G+1$. Let $Y_{new} = Y_{n+1}$. With loss $L$ right-continuous w.r.t. $\lambda$ and bounded in $[0,1]$ and $L(\cdot, \lambda_{max}) \leq \alpha$, the estimator $\hat{\lambda}_Z$ returned by Algorithm 4.1 satisfies*

$$\mathbb{E}\big[L(Y_{\text{new}}, \hat{\lambda}_Z)\big] \leq \alpha.$$

*Proof.* First relate the fresh outcome $Y_{\text{new}}$ to the next-round pseudo-outcomes $\{Z_j^{G+1}\}_{j=1}^K$:

$$\mathbb{E}\big[L(Y_{\text{new}}, \hat{\lambda}_Z)\big] = \mathbb{E}\Big[\mathbb{E}\big[L(Y_{\text{new}}, \hat{\lambda}_Z) \,|\, \{Z_j^g\}_{j,g=1}^{K,G}\big]\Big]$$
$$= \mathbb{E}\Big[\mathbb{E}\big[L(Z_{j'}^{G+1}, \hat{\lambda}_Z) \,|\, \{Z_j^g\}_{j,g=1}^{K,G}\big]\Big], \; (j' = 1, \ldots, K)(\text{By Lemma 4.1})$$
$$= \mathbb{E}\Big[\mathbb{E}\big[\tfrac{1}{K}\sum_{j=1}^K L(Z_j^{G+1}, \hat{\lambda}_Z) \,|\, \{Z_j^g\}_{j,g=1}^{K,G}\big]\Big]$$
$$= \mathbb{E}\Big[\tfrac{1}{K}\sum_{j=1}^K L(Z_j^{G+1}, \hat{\lambda}_Z)\Big].$$

Define

$$\hat{\lambda}_Z' = \inf\Big\{\lambda : \tfrac{1}{(G+1)K}\sum_{g=1}^{G+1}\sum_{j=1}^K L(\mathbf{Z}_j^g, \lambda) \leq \alpha\Big\}.$$

then $\hat{\lambda}_Z \geq \hat{\lambda}_Z'$, thus

$$L(Z_j^{G+1}, \hat{\lambda}_Z) \; \leq \; L(Z_j^{G+1}, \hat{\lambda}_Z'), \; (j = 1, \ldots, K)$$

because $L(\cdot, \lambda)$ is decreasing with respect to $\lambda$. Hence

$$\mathbb{E}\Big[\tfrac{1}{K}\sum_{j=1}^K L(Z_j^{G+1}, \hat{\lambda}_Z)\Big] \; \leq \; \mathbb{E}\Big[\tfrac{1}{K}\sum_{j=1}^K L(Z_j^{G+1}, \hat{\lambda}_Z')\Big].$$

Exchangeability of the $G+1$ blocks implies

$$\mathbb{E}\big[\tfrac{1}{K}\sum_{j=1}^{K}L(Z_j^{G+1},\hat{\lambda}_Z')\big] = \mathbb{E}\Big[\mathbb{E}\big[\tfrac{1}{K}\sum_{j=1}^{K}L(Z_j^{G+1},\hat{\lambda}_Z')\,|\,\{Z_j^g\}_{j,g=1}^{K,G+1}\big]\Big]$$

$$= \mathbb{E}\Big[\mathbb{E}\big[\tfrac{1}{G+1}\sum_{g=1}^{G+1}\tfrac{1}{K}\sum_{j=1}^{K}L(Z_j^g,\hat{\lambda}_Z')\,|\,\{Z_j^g\}_{j,g=1}^{K,G+1}\big]\Big]$$

(By exchangeability of $\{\{Z_j^g\}_{j=1}^{K}\}_{g=1}^{G+1}$)

$$= \mathbb{E}\big[\tfrac{1}{G+1}\sum_{g=1}^{G+1}\tfrac{1}{K}\sum_{j=1}^{K}L(Z_j^g,\hat{\lambda}_Z')\big]$$

$$\leq \alpha, \ \text{(By definition of } \hat{\lambda}_Z')$$

establishing the desired risk bound. $\qquad\square$

**Theorem 4.3** (Finite-sample RBWA-CRC). *Assume $\{B_g\}_{g=1}^{G+1}$ are i.i.d and $\{Y_{g,1},\dots,Y_{g,I}\}$ are exchangeable for $g=1,2,\dots,G+1$. Let $Y_{new} = Y_{n+1} = Y_{G+1,1}$. With loss $L$ right-continuous w.r.t. $\lambda$ and bounded in $[0,1]$ and $L(\cdot,\lambda_{max}) \leq \alpha$, the estimator $\hat{\lambda}_p$ returned by Algorithm 4.2 satisfies*

$$\mathbb{E}\big[L(Y_{\text{new}},\hat{\lambda}_p)\big] \leq \alpha.$$

*Proof.* Imagine that we have pseudo-batch $B_{G+1} = \{Y_{G+1,i}\}_{i=1}^{I}$ and pseudo-sample $p_{G+1} \sim \mathcal{P}_S$ which is independent of $\{p_g\}_{g=1}^{G}$ and $\{B_g\}_{g=1}^{G+1}$. Now

$$\mathbb{E}\big[L(Y_{\text{new}},\hat{\lambda}_p)\big] = \mathbb{E}\Big[\mathbb{E}\big[L(Y_{G+1,1},\hat{\lambda}_p)\,|\,\{(B_g,p_g)\}_{g=1}^{G},p_{G+1}\big]\Big]$$

$$= \mathbb{E}\Big[\mathbb{E}\big[L(Y_{G+1,i},\hat{\lambda}_p)\,|\,\{(B_g,p_g)\}_{g=1}^{G},p_{G+1}\big]\Big], \ (i=1,\dots,I)$$

(By exchangeability of $\{Y_{G+1,i}\}_{i=1}^{I}$)

$$= \mathbb{E}\Big[\mathbb{E}\big[\sum_{i=1}^{I}p_{G+1,i}L(Y_{G+1,i},\hat{\lambda}_p)\,|\,\{(B_g,p_g)\}_{g=1}^{G},p_{G+1}\big]$$

$$\text{(This line follows from } \sum_{i=1}^{I}p_{G+1,i}=1)\Big]$$

$$= \mathbb{E}\Big[\sum_{i=1}^{I}p_{G+1,i}L(Y_{G+1,i},\hat{\lambda}_p)\Big]$$

$$= \mathbb{E}\Big[L_{G+1}(\hat{\lambda}_p)\Big]$$

Notice that $\{L_g\}_{g=1}^{G+1}$ satisfy the assumption of Theorem 1 in "*Conformal Risk Control*", thus our theorem holds.

Define

$$\hat{\lambda}_p' = \inf\Big\{\lambda : \tfrac{1}{G+1}\sum_{g=1}^{G+1}L_g(\lambda) \leq \alpha\Big\}.$$

then $\hat{\lambda}_p \geq \hat{\lambda}_p'$, thus

$$L(Y_{G+1,i},\hat{\lambda}_p) \ \leq \ L(Y_{G+1,i},\hat{\lambda}_p'), \ (i=1,2,\dots,I)$$

because $L(\cdot,\lambda)$ is decreasing with respect to $\lambda$. Hence

$$\mathbb{E}\Big[L_{G+1}(\hat{\lambda}_p)\Big] \ \leq \ \mathbb{E}\Big[L_{G+1}(\hat{\lambda}_p')\Big].$$

Exchangeability of the $G + 1$ blocks implies

$$\mathbb{E}\big[L_{G+1}(\hat{\lambda}'_p)\big] = \mathbb{E}\Big[\mathbb{E}\big[L_{G+1}(\hat{\lambda}'_p)\,\big|\,\{(B_g, p_g)\}_{g=1}^{G+1}\big]\Big]$$

$$= \mathbb{E}\Big[\mathbb{E}\big[\tfrac{1}{G+1}\sum_{g=1}^{G+1} L_g(\hat{\lambda}'_p)\,\big|\,\{(B_g, p_g)\}_{g=1}^{G+1}\big]\Big] \ \text{(By exchangeability of } \{(B_g, p_g)\}_{g=1}^{G+1})$$

$$= \mathbb{E}\big[\tfrac{1}{G+1}\sum_{g=1}^{G+1} L_g(\hat{\lambda}'_p)\big]$$

$$\leq \alpha, \ \text{(By definition of } \hat{\lambda}'_p)$$

$\square$

*

*Proof.* Let $\{Z_j^g\}_{j \leq K,\, g \leq G}$ be the calibration replicates, and let $\{Z_j^{G+1}\}_{j \leq K}$ denote hypothetical replicates from a future batch $G+1$. Conditioning on calibration batches and using Lemma 4.1,

$$\mathbb{E}\Big[\ell_{\hat{\lambda},\beta}(Y_{\text{new}})\,\Big|\,\{Z_j^g\}\Big] = \mathbb{E}\big[\bar{\ell}_{\hat{\lambda}}(G+1, j)\,\big|\,\{Z_j^g\}\big], \quad j = 1, \dots, K,$$

hence

$$\mathbb{E}\big[\ell_{\hat{\lambda},\beta}(Y_{\text{new}})\big] = \mathbb{E}\left[\frac{1}{K}\sum_{j=1}^{K} \bar{\ell}_{\hat{\lambda}}(G+1, j)\right].$$

By calibration, $\widehat{R}_{BB}(\hat{\lambda}) + \tfrac{1}{G} \leq \alpha$ implies $\widehat{R}_{BB}(\hat{\lambda}) \leq \alpha - \tfrac{1}{G}$. Since each $\bar{\ell}_\lambda(g, j) \in [0, 1]$, the augmented $(G+1)$-batch average obeys

$$\frac{1}{(G+1)K}\left(\sum_{g=1}^{G}\sum_{j=1}^{K} \bar{\ell}_{\hat{\lambda}}(g, j) + \sum_{j=1}^{K} \bar{\ell}_{\hat{\lambda}}(G+1, j)\right) \leq \frac{G}{G+1}\left(\alpha - \tfrac{1}{G}\right) + \frac{1}{G+1} \leq \alpha.$$

Taking expectations and using exchangeability across batches yields $\mathbb{E}[\ell_{\hat{\lambda},\beta}(Y_{\text{new}})] \leq \alpha$. $\square$

Appendix: Proofs for §4.3

**Theorem 4.4** (RBWA moments: unbiased smoothing, variance dial, and anti-concentration). *Let $p_g \sim \text{Dirichlet}(\eta\mathbf{1})$ with $\eta > 0$ and set $\kappa := I\eta$. For any fixed $\lambda$ and any bounded losses $\{\ell_{g,i}(\lambda)\}_{i=1}^{I} \subset [0, 1]$:*

*(a) Unbiasedness:* $\mathbb{E}\big[L_g(\lambda)\,|\,\ell\big] = \mu(\lambda)$.

*(b) Variance dial:* $\text{Var}\big(L_g(\lambda)\,|\,\ell\big) = \text{Var}_{\text{emp}}(\ell_g(\lambda))/(\kappa + 1)$. *Thus, for any $t > 0$,*

$$\Pr\left(|L_g - \mu| \geq t\,|\,\ell\right) \leq \frac{\text{Var}_{\text{emp}}(\ell_g)}{(\kappa + 1)t^2}, \qquad \Pr\left(L_g \geq \mu + t\,|\,\ell\right) \leq \frac{\text{Var}_{\text{emp}}(\ell_g)}{\text{Var}_{\text{emp}}(\ell_g) + (\kappa + 1)t^2}.$$

*(c) Anti-concentration: if $(\ell_1(\lambda), \dots, \ell_I(\lambda))$ is not constant, then $L_g(\lambda)$ has no atoms $(\Pr(L_g = t\,|\,\ell) = 0$ for all $t)$, hence threshold ties caused by discrete lattice values disappear.*

*Proof.* (a)–(b) For symmetric Dirichlet with total mass $\kappa = I\eta$, $\mathbb{E}[p_{g,i}] = 1/I$, $\text{Var}(p_{g,i}) = \frac{I-1}{I^2(\kappa+1)}$, $\text{Cov}(p_{g,i}, p_{g,j}) = -\frac{1}{I^2(\kappa+1)}$ for $i \neq j$. Hence $\mathbb{E}[L_g\,|\,\ell] = \sum_i \mathbb{E}[p_{g,i}]\ell_i = \mu$ and

$$\text{Var}(L_g\,|\,\ell) = \sum_i \text{Var}(p_{g,i})\ell_i^2 + 2\sum_{i<j}\text{Cov}(p_{g,i}, p_{g,j})\ell_i\ell_j = \frac{\text{Var}_{\text{emp}}(\ell)}{\kappa + 1}.$$

Chebyshev's Inequality and Cantelli's Inequality yield the displayed tail bounds.

(c) The Dirichlet law has a continuous density on the simplex interior (for parameters $> 0$). The linear map $f(p) = \sum_i p_i \ell_i$ is non-constant when the $\ell_i$ are not all equal; its level set $\{p : f(p) = t\}$ is a codimension-1 slice of the simplex and has Lebesgue measure zero. Therefore $\Pr(L_g = t\,|\,\ell) = 0$. $\square$

| ID | Benchmark (split) | #Q | Para $P$ | Ans $N$ | Mix (N/E/Z) | Entropy $\tau$ |
|----|-------------------|-----|----------|---------|-------------|----------------|
| C1 | ASQA (dev) | 60 | 10 | 150 | (.75/.00/.25) | 0.90 |
| C2 | NQ-Open (val) | 60 | 6 | 16 | (.67/.00/.33) | 0.86 |
| C3 | HotpotQA (val) | 60 | 10 | 100 | (.60/.00/.40) | 0.86 |
| C4 | AmbigQA (dev) | 60 | 10 | 150 | (.75/.00/.25) | 0.86 |
| C5 | AmbigQA (dev) (ablation: decoding entropy) | 40 | 10 | 150 | (.75/.00/.25) | 0.86 |
| C6 | NQ-Open (val) (ablation: vendor/model) | 60 | 6 | 16 | (.67/.00/.33) | 0.86 |

Table 4: Benchmarks and per-item sampling settings used in the hallucination study. The mix column shows (normal/enforced/noise).

**Theorem 4.5** (RBWA calibration CLT under precision stabilization). *Fix a $\lambda$. Assume batches are i.i.d., losses are bounded in $[0, 1]$. Let $p_g \sim \mathrm{Dirichlet}(\mathbf{1})$. Let*

$$\mu(L) = \mathbb{E}[\mu_g], \ \mathrm{Var}(L) = \frac{\mathbb{E}[\mathrm{Var}_{\mathrm{emp}}(\ell_g)]}{+1} + \mathrm{Var}(\mu_g)$$

*they are well-defined because $\{\ell_g\}_{g=1}^G$ are i.i.d.. Assume $\mathrm{Var}(\ell)$ is finite. Then*

$$\sqrt{G}\left(\bar{L}_G - \mu(L)\right) \Rightarrow \mathcal{N}(0, \mathrm{Var}(L)), \qquad \bar{L}_G = \frac{1}{G}\sum_{g=1}^{G} L_g$$

*as $G \to \infty$.*

*Proof. Conditional moments.* By Theorem 4.1, $\mathbb{E}[L_g \mid \ell] = \mu(\lambda)$ and $\mathrm{Var}(L_g \mid \ell) = \mathrm{Var}_{\mathrm{emp}}(\ell(\lambda))/(\kappa + 1) \to \mathrm{Var}_{\mathrm{emp}}(\ell(\lambda))/(\chi + 1)$ in probability.

*Triangular-array CLT.* For fixed $\lambda$, the $L_g(\lambda)$ are independent, uniformly bounded, and have asymptotically constant variance. The Lindeberg–Feller CLT applies, giving

$$\sqrt{G}\left(\bar{L}_G(\lambda) - \mathbb{E}[L_g(\lambda)]\right) \Rightarrow \mathcal{N}(0, \ \mathrm{Var}_{\mathrm{emp}}(\ell(\lambda))/(\chi + 1)).$$

Since $\mathbb{E}[L_g(\lambda)] = \mu(\lambda)$, Slutsky's lemma yields the stated limit. $\qquad\square$

# B EXPERIMENT

## B.1 APPENDIX: HALLUCINATION EXPERIMENT SETTINGS AND CONFIGURATIONS

**Scope and outputs.** For each question we generate a response cloud, compute **Factuality Severity** $= 1 - \max_{r \in \mathrm{refs}} \mathrm{BERTScore}\text{-}F1(a, r)$. All runs are seeded and logged to timestamped, self-describing CSVs: a per-*answer* file (scores, margins, types, decoding knobs) and a per-*run* file (dataset/split, sample counts, model/provider, seeds, thresholds, and paths). Together model IDs are normalized to serverless fallbacks to avoid availability regressions.

**Benchmarks and roles.** We evaluate across four core datasets—ASQA (dev), NQ-Open (validation), HotpotQA (validation), AmbigQA (dev)—plus two ablations that stress decoding entropy and vendor/model choice. Each configuration fixes decoding knobs and the normal/enforced/noise mix, while paraphrasing a canonical gold to reduce aliasing of surface forms.

*Shared knobs:* alias-normalization for Together; n_per_call=5; rate-limit $\approx 0.8$s; severity mix weight logged; seeds: C1=42, C2=7, C3=11, C4=23, C5=23, C6=8.

**Example prompts (verbatim, used in data generation).** We use minimal, auditable prompts. For *paraphrasing* the canonical gold: *System:* "You rewrite text. Output a succinct standalone paraphrase." *User:* "Paraphrase the following answer in different wording, preserving the exact meaning and factual content. Keep it concise and standalone. Avoid hedging, qualifiers, or extra details. **Answer:** {*gold*}." For *normal answers:* *System:* "Answer the question with the canonical short answer first; then add at most one brief justification. Be concise." *User:* {*question*}. For *enforced canonical answers:* *System:* "Answer with the canonical short answer first; then a single, concrete supporting detail. Avoid aliasing, avoid hedging, avoid contradictory statements." *User:* {*question*}. (Noise/outlier strings are programmatically injected: gibberish, off-topic, fabricated citations, prompt-injection strings, contradictions, emoji floods, and multilingual snippets.)

| ID | Provider | Model | Temp | Top-$p$ | MaxTok | Embed | BERTScore |
|----|----------|-------|------|---------|--------|-------|-----------|
| C1 | Together | Llama-3.3-70B-Instr. Turbo | 1.3 | 1.0 | 256 | MiniLM-L6-v2 | RoBERTa-large |
| C2 | OpenAI | gpt-4o-mini | 0.1 | 1.0 | 96 | MiniLM-L6-v2 | RoBERTa-large |
| C3 | Together | Mixtral-8x7B-Instr. v0.3 | 1.2 | 1.0 | 256 | MiniLM-L6-v2 | RoBERTa-large |
| C4 | Together | Llama-3.1-8B-Instr. Turbo | 0.7 | 0.9 | 256 | MiniLM-L6-v2 | RoBERTa-large |
| C5 | Together | Llama-3.1-8B-Instr. Turbo | 1.3 | 1.0 | 256 | MiniLM-L6-v2 | RoBERTa-large |
| C6 | Together | Llama-3.1-8B-Instr. Turbo | 0.1 | 1.0 | 96 | MiniLM-L6-v2 | RoBERTa-large |

Table 5: Provider/decoding and measurement settings, linked by **ID** to Table 4.

## B.2 LLM-AS-A-JUDGE: IMPLEMENTATION DETAILS

We use an LLM as a rubric-based grader for short QA answers, producing a continuous 0–100 direct-assessment score on the answer head (same head definition as in the main text). Scores are later normalized and mapped to severity via $\mathrm{JS}(a) = 1 - \mathrm{J}(a)/100$.

### JUDGE MODEL AND PARAMETERS

- **Primary model:** `meta-llama/Meta-Llama-3.1-8B-Instruct-Turbo` (Together).
- **Alternative:** `gpt-4o-mini` (OpenAI).
- **Decoding:** temperature $= 0.2$, top_p $= 1.0$, max_tokens $= 200$.
- **Output scale:** single overall score in $[0, 100]$ with one decimal.
- **Seed/reproducibility:** fixed RNG seed $= 7$ to stabilize few-shot ordering.
- **Rubric weights:** correctness 0.60, faithfulness 0.20, completeness 0.15, clarity 0.05 (weights sum to 1.0).

### B.2.1 WHAT IS SCORED

For each item, the judge sees: (i) the question text $q$, (ii) a compact reference set $\mathcal{R}_q$ (concise paraphrases of acceptable answers), and (iii) the *answer head* of the candidate $a$ (first sentence or `Final:` field, truncated to $\leq 16$ tokens). This reduces style/length effects and prevents rationales from masking factual errors.

We use an aspect rubric with fixed weights:

1. **Correctness** (0.60): factual accuracy and semantic match to $q$ and $\mathcal{R}_q$.
2. **Faithfulness** (0.20): no contradictions or hallucinations relative to $\mathcal{R}_q$.
3. **Completeness** (0.15): covers the essential point(s) asked by $q$.
4. **Clarity** (0.05): concise and unambiguous phrasing.

To reduce score compression, the prompt embeds 3 few-shot *anchors* (low/mid/high) that illustrate how the 0–100 scale should be used.

### B.2.2 PROMPT TEMPLATE (VERBATIM)

We employ a compact, instruction-only template that requires strict JSON output. Placeholders in {braces} are filled per example.

```
You are a meticulous grader for short QA responses.
Score each candidate on a 0{100 scale with ONE decimal place (e.g., 81.7).
Be strict: wrong facts should sharply reduce the score. If no references
are provided, use general knowledge.

Rubric (weights sum to 1.0):
- Correctness ({w_correctness}):

factual accuracy & semantic match to the question and references.
- Faithfulness ({w_faithfulness}):

no hallucinations or contradictions vs. references.
- Completeness ({w_completeness}):
```

```
covers essential point(s) requested.
- Clarity ({w_clarity}):

concise, unambiguous wording.

Scoring examples (anchors):
{anchors_block}

Return ONLY a compact JSON object with keys "score" and "subscores";
"subscores" must contain "correctness", "faithfulness", "completeness",
"clarity". Example:
{"score": 88.6, "subscores": {"correctness": 56.0, "faithfulness": 17.6,
 "completeness": 11.0, "clarity": 4.0}}

Now grade this:
Question: {question}
References: {references}   # short list or "N/A"
Candidate: {candidate}     # answer head only
```

### B.2.3 ANCHOR EXAMPLES (VERBATIM)

We rotate/shuffle three anchors (by a fixed seed) to calibrate the scale.

```
- Q: "Who wrote 'Pride and Prejudice'?"
  A: "Pride and Prejudice was written by Jane Austen." -> 95.0 (exact)
- Q: "Who wrote 'Pride and Prejudice'?"
  A: "It's probably Charles Dickens, I guess." -> 15.0 (incorrect; hedging)
- Q: "Capital of Australia?"
  A: "Canberra. It's not Sydney or Melbourne." -> 90.0 (precise; disambiguates)
```

### B.2.4 NORMALIZATION AND SEVERITY

The judge produces $J(a) \in [0, 100]$; we compute $J_{norm}(a) = J(a)/100$ and

$$JS(a) = 1 - J_{norm}(a) \in [0, 1].$$

Low JS indicates high factual alignment; values near 1 flag deviation via error, omission, or contradiction.

### B.3 APPENDIX (FOR §4.2): IMPLEMENTATION AND CROSS-VALIDATION DETAILS

**Data and artifacts (two policy scores).** For each question we materialize a *self-consistency queue* by sampling a small cloud of responses or judge rationales under fixed decoding knobs. Each element is embedded with a single encoder (unit-norm rows), and we log two separable signals per item $y$: (i) a *label-free* geometry score $E(y) \in [0, 1]$ from Gram row energies (Eq. 2.4; aggregated and projected as defined); and (ii) an *offline* factuality-severity flag $q_\beta(y) \in [0, 1]$ drawn from references or rubric-graded scores (e.g., $FS = 1 - BERTScore\text{-}F1$, or $JS = 1 - J/100$ from an LLM judge). When using the judge as an *online* policy, we also record the normalized judge score $J_{norm}(y) = J(y)/100 \in [0, 1]$ along with subscores (correctness, faithfulness, completeness, clarity) from a fixed rubric/anchor prompt. The responses-level CSV stores per-item $E$ summaries (e.g., queue median), $q_\beta$, and (optionally) $J_{norm}$ with subscores; a parameters-level CSV records dataset/model/seed/knobs for reproducibility.

| Policy score $Q$ (definition) | Action (gate) | Key strengths |
|---|---|---|
| $Q_E$: $Q_E(y) = E(y)$ (Eq. 2.4) | Accept on high consensus | Gram geometry; centroid coupling; statistically traceable |
| $Q_G$: $Q_G(y) = J_{norm}(y)$ (§4.1) | Accept on high judge score | Direct control of judge pipeline; reliability gains; rubric-aligned |

Table 6: Compact summary of the two instantiations of the $Q$-policy.

We instantiate the policy-first loss of Eq. (4.2) with *either* geometry or judge as $Q$:

$$\mathcal{L}_E(y, \lambda) = \mathbf{1}\{E(y) \geq \lambda\} q_\beta(y), \qquad \mathcal{L}_G(y, \lambda) = \mathbf{1}\{J_{norm}(y) \geq \lambda\} q_\beta(y).$$

Both are *monotone in* $\lambda$: as $\lambda$ increases, the system accepts fewer items, so the loss cannot increase. Interpretation is identical: we incur loss only when we *act* (accept) and the item is *bad* (unfactual according to $q_\beta$).

Normalizing $E$ and $J_{\text{norm}}$ to $[0, 1]$ renders $\lambda$ dimensionless and comparable across folds/days. For robustness we may Huberize/clip $q_\beta$ to dampen tails; the actuator never reads $q_\beta$ online. (Judge scoring, normalization, and rubric details appear in §5.2; severity JS in Eq. (4.2).)

To assess generalization at the "new question" granularity, we adopt grouped CV with disjoint question blocks per fold. On calibration folds we treat pairs $(Q_i, q_{\beta,i})$ as exchangeable—where $Q_i \in \{E_i, J_{\text{norm},i}\}$ depending on the policy—and apply CRC to select a single global threshold *per policy*, $\hat{\lambda}_E(\alpha)$ and $\hat{\lambda}_G(\alpha)$. We use two compute-aware calibrators: BB–CRC (batched bootstrap reuse) and RBWA–CRC (randomized simplex weights). Both operate on the same bounded, monotone losses and differ only in how they stabilize the empirical risk curve. In BB–CRC, the smallest right-open threshold satisfying the bias-corrected constraint

$$\frac{1}{(G+1)K} \sum_{g=1}^{G} \sum_{j=1}^{K} \mathcal{L}\big(Z_j^{(g)}, \lambda\big) \; + \; \frac{1}{G+1} \; \leq \; \alpha$$

is returned as $\hat{\lambda}$; RBWA–CRC replaces per-replicate sums by per-batch randomized weighted averages $\sum_i p_{g,i} \mathcal{L}(Y_{g,i}, \lambda)$ with $p_g \sim \text{Dirichlet}(\eta \mathbf{1})$, yielding an unbiased smoother with a one-knob variance dial. The calibrator only sees $(Q, q_\beta)$ pairs; no labels or logits are needed at deployment time. (Formal CRC details and the compute-aware variants are in §3.)

We calibrate *one threshold per policy* on a scalar, deployment-time score $Q$: $Q_E = E$ (label-free Gram energy) or $Q_G = J_{\text{norm}}$ (LLM-as-Judge). Both share the same policy-first actuator $\mathbf{1}\{Q \geq \hat{\lambda}\}$ and the same calibration-only severity $q_\beta$, yielding a single-knob control that transfers unchanged to production. Geometry offers statistical traceability and interpretability via auditable batch consensus, while the judge policy directly steers judge-driven pipelines and can improve their reliability under a frozen rubric—all without requiring ground truth at runtime. (See §3 for guarantees and §5.2 for judge implementation.)

## C  BASELINE IMPLEMENTATION DETAILS

Each benchmark dataframe row contains at least: `severity_f1` (unfiltered factuality severity; lower is better) and a judge score (`LLMJUDGE_score_norm` $\in [0, 1]$, or `LLMJUDGE_score`/100). Other columns (question, model, provider, etc.) are logged but unused by the actuator. FS is defined as $FS = 1 - \text{BERTScore-F1}(\text{answer head}, \text{reference})$ with head length $\leq 16$ tokens.

We use the policy-first loss and gate (Eq. (4.1), (5.3)):

$$L(y, \lambda) = a_\lambda(Q(y)) \cdot q_\beta(y), \qquad a_\lambda(u) = \mathbf{1}\{u \geq \lambda\}.$$

At deployment we compute $Q(y)$ and apply $a_{\hat{\lambda}}(Q(y))$; $q_\beta$ is calibration-only. This yields a bound on acted-while-bad intensity $E[L(Y_{\text{new}}, \hat{\lambda})] \leq \alpha$ in finite samples for the CRC modes.

*G-Eval-N* (G-Eval Naive): $Q = J_{\text{norm}}$. We evaluate a fixed list of thresholds $\lambda \in \{0.99, 0.95, 0.90, 0.85, 0.80\}$. For fairness to CRC plots, we display the corresponding cells beside $\alpha \in \{0.01, \dots, 0.20\}$ (visual alignment only; no guarantees). *G-Eval-CRC* (G-Eval Risk Control): $Q = J_{\text{norm}}$. We calibrate a single $\hat{\lambda}(\alpha)$ per setting using **BB-CRC** (Alg. 4.1, Thm. 4.2) and deploy the same hard gate. *Grand-CRC* (Grand Risk Control): $Q = E$ (centered-Gram energy, $[0, 1]$ after normalization). We calibrate $\hat{\lambda}(\alpha)$ using **BB-CRC** (Alg. 4.2, Thm. 4.3) with Dirichlet weights at fixed precision, then deploy the same hard gate.

Given a dataframe $D$ with columns $\{\texttt{severity\_f1}, \texttt{LLMJUDGE\_score\_norm}\}$:

1. For each threshold $\lambda$ (fixed in G-Eval-N or calibrated $\hat{\lambda}(\alpha)$ in CRC modes), define shipped mask $M = \{J_{\text{norm}} \geq \lambda\}$ (for judge modes) or $M = \{E \geq \lambda\}$ (for Grand-RC).

2. Compute $\text{FS}_{\text{shipped}} = \text{mean}(\texttt{severity\_f1}[M])$, $\text{FS}_{\text{unshipped}} = \text{mean}(\texttt{severity\_f1}[\neg M])$.

3. Report FS-reduction$(\%) = 100 \cdot \big(1 - \text{FS}_{\text{shipped}}/\text{FS}_{\text{unshipped}}\big)$ and the acceptance rate $|M|/|D|$.

When either group is empty we emit `NaN` and exclude from the aggregate.

## D  MORE EXPERIMENT RESULTS

The six panels span four QA regimes—AmbigQA (aliases/answer sets), NQ-Open (single-hop), HotpotQA (multi-hop), and ASQA (under-specification)—plus two controlled ablations designed to probe robustness: a high-entropy decoding setting (AMBIGQA-ENT) and a vendor/model swap on NQ-Open. The short codes in Table 7 bind figure titles to the exact CSV artifacts used for reproducibility and align with our FS-on-answer-head measurement.

Table 7: **Benchmark mapping used in the six-panel comparisons.** Short codes are the compact labels used in figure titles. For JudgeQ CSVs, the same names appear with the suffix `__judged`.

| Panel | Short code | CSV `dataset_name` |
|---|---|---|
| 1 | AMBIGQA-ENT | `ambigqa__llama8b__hiT_ablation_entropy__ns40_responses` |
| 2 | AMBIGQA | `ambigqa__llama8b__midT_ns60_responses` |
| 3 | ASQA | `asqa__llama70b__hiT__ns60_responses` |
| 4 | HOTPOTQA | `hotpot_mixtral8x7b__hiT_noise40__ns60_responses` |
| 5 | NQ-OPEN | `nq__gpt4omini__loT_light__ns60_responses` |
| 6 | NQ-OPEN-VEND | `nq__llama8b__loT_ablation_vendor__ns60_responses` |

**Reading the six-panel plots.** For each budget $\alpha \in \{0.01, \ldots, 0.20\}$, the (calibrated) actuator $a_{\hat{\lambda}}$ partitions candidates into *Unshipped* and *Shipped*; bars report mean for each group (lower is better). Panel titles use the short codes above; legends are placed outside the axes to preserve title visibility. The gate and metric are fixed; only the policy score changes between the next two figures.

$Q_E$ **(Gram geometry) results.** Across all six panels, the shipped sets exhibit a large drop for every $\alpha$, including the entropy stress test (panel 1) and the vendor swap (panel 6), indicating stability to decoding noise and provider/model variation. Aggregated over panels, the geometry policy sustains high reductions as $\alpha$ grows (e.g., $97.9\% \rightarrow 86.0\%$ from $\alpha{=}0.01$ to $0.20$ in Table 8), consistent with consensus-seeking acceptance in centered Gram space.

**Switching the policy to $Q_J$.** We now hold the actuator and calibration protocol fixed and replace the online score with a rubric-normalized judge ($Q_J$), so differences isolate the *policy signal* rather than changes in gating or measurement.

$Q_J$ **(LLM-as-judge) results.** The judge policy also lifts factuality across panels but shows a stronger dependence on $\alpha$ (and mild task-to-task variation), which is compatible with rubric/style sensitivity; CRC turns its threshold into a measurable one-knob control with finite-sample validity. In the compact summary (Table 8), the FS reduction moves from $98.9\%$ at $\alpha{=}0.01$ to $46.5\%$ at $\alpha{=}0.20$.

**Compact cross-policy summary.** Table 8 aggregates the six-panel plots by reporting ($_{\text{unshipped}}, _{\text{shipped}}$) and the percentage reduction at each $\alpha$ for both policies. Geometry maintains uniformly lower shipped severity across budgets, while the judge policy is competitive at tight budgets and provides an interpretable baseline for a rubric-driven pipeline under the same actuator.

Table 8: **Compact FS reduction across $\alpha$.** For each policy (GramQ, JudgeQ) and $\alpha$, we report $\text{FS}_{\text{unshipped}}$, $\text{FS}_{\text{shipped}}$, and the percentage reduction from Unshipped to Shipped (higher is better). FS follows the paper's definition $FS = 1 - \text{BERTScore-F1}(\text{answer head}, \text{reference})$.

| Policy | $\alpha$ | $\text{FS}_{\text{unshipped}}$ | $\text{FS}_{\text{shipped}}$ | FS reduction (%) |
|---|---|---|---|---|
| GramQ | 0.01 | 0.892 | 0.019 | 97.9 |
| GramQ | 0.05 | 0.892 | 0.030 | 96.7 |
| GramQ | 0.10 | 0.885 | 0.057 | 93.6 |
| GramQ | 0.15 | 0.874 | 0.093 | 89.4 |
| GramQ | 0.20 | 0.859 | 0.120 | 86.0 |
| JudgeQ | 0.01 | 0.903 | 0.010 | 98.9 |
| JudgeQ | 0.05 | 0.904 | 0.197 | 78.3 |
| JudgeQ | 0.10 | 0.905 | 0.314 | 65.3 |
| JudgeQ | 0.15 | 0.905 | 0.408 | 55.0 |
| JudgeQ | 0.20 | 0.907 | 0.485 | 46.5 |

**Calibration quality and stability.** Table 9 summarizes empirical risk and threshold stability for CRC variants. All three keep acted-while-bad risk near or below the budget, while RBWA tracks $\alpha$ most closely and reduces the standard error of the calibrated threshold: e.g., at $\alpha{=}0.15$ the empirical risks are 0.039 (BB-CRC), 0.039 (CRC), 0.138 (RBWA) with $\text{SE}(\hat{\lambda})$ of $7.46 \times 10^{-4}$, $5.79 \times 10^{-4}$, and $2.89 \times 10^{-4}$, respectively; at $\alpha{=}0.05$, $\text{SE}(\hat{\lambda})$ falls from $\approx 1.3 \times 10^{-3}$ (BB-CRC/CRC) to $4.61 \times 10^{-4}$ (RBWA). These patterns match the smoothing/anti-concentration analysis for RBWA and the bootstrap reuse in BB-CRC.

Table 9: **Calibration summary across** $\alpha$. Empirical risk and its standard error (SE) for each method, together with the **Stability** (SE of $\lambda$).

| Method | $\alpha$ | Empirical risk | Risk SE | Stability (SE of $\lambda$) |
|---|---|---|---|---|
| BB-CRC | 0.01 | 0.012 | 0.012 | 0.001299 |
| BB-CRC | 0.05 | 0.012 | 0.012 | 0.001299 |
| BB-CRC | 0.10 | 0.026 | 0.018 | 0.000375 |
| BB-CRC | 0.15 | 0.039 | 0.022 | 0.000746 |
| BB-CRC | 0.20 | 0.062 | 0.030 | 0.000194 |
| CRC | 0.01 | 0.012 | 0.012 | 0.001321 |
| CRC | 0.05 | 0.012 | 0.012 | 0.001321 |
| CRC | 0.10 | 0.026 | 0.018 | 0.000419 |
| CRC | 0.15 | 0.039 | 0.022 | 0.000579 |
| CRC | 0.20 | 0.062 | 0.030 | 0.000248 |
| RBWA | 0.01 | 0.000 | 0.000 | 0.000000 |
| RBWA | 0.05 | 0.026 | 0.018 | 0.000461 |
| RBWA | 0.10 | 0.074 | 0.031 | 0.000189 |
| RBWA | 0.15 | 0.138 | 0.042 | 0.000289 |
| RBWA | 0.20 | 0.171 | 0.045 | 0.000236 |

**Takeaway.** Across heterogeneous QA regimes and stress tests, a single calibrated gate converts variability into validity; $Q_E$ provides a provider-agnostic consensus signal with uniform gains, while $Q_J$ with CRC yields a deployable judge pipeline with a measurable, risk-tracked knob.

