# OpenReview forum: "Taming Variability: Randomized and Bootstrapped Conformal Risk Control for LLMs"
_ICLR.cc/2026/Conference — ICLR 2026 Conference Withdrawn Submission_

### Official Review · Reviewer_8S5h · 2025-10-30

**Soundness:** 2
**Presentation:** 2
**Contribution:** 2
**Rating:** 4
**Confidence:** 2

**Summary:**

This paper proposes a Conformal Actuator (CA) framework for black-box LLM deployment. The method uses Conformal Risk Control (CRC) to enforce a user-specified risk budget at the API boundary, routing responses via a monotone gate (ship/abstain/regenerate/escalate). Two calibration variants, BB-CRC and RBWA-CRC, aim to improve efficiency and threshold stability. A Gram-geometry–based score is introduced as a label-free uncertainty signal. Experiments on several QA datasets show reductions in factuality errors.

**Strengths:**

The problem of controlling the reliability and variability of black-box LLMs is timely and relevant. The design of a single calibrated actuator is conceptually appealing: one risk knob exposed to practitioners with finite-sample guarantees. The separation between an offline risk label and an online label-free score is elegant. The computational variants are technically sound and target practical deployment costs. Experiments show consistent factuality improvements across datasets and providers, and the Gram-based score is an interesting idea for model-agnostic uncertainty.

**Weaknesses:**

The paper lacks conceptual clarity on what risk is actually controlled. Hallucination is not formally defined. What I understand is that hallucination refers to the model producing wrong or unsupported facts. CRC, however, controls miscoverage risk, i.e., the probability that the true value is excluded from the accepted set under a chosen loss. This is not the same as factual correctness. Conformal prediction provides uncertainty calibration, not correctness guarantees. The manuscript repeatedly suggests that CRC “mitigates hallucination”, but the connection is indirect and not theoretically justified. If the intended claim is that CRC reduces overconfident hallucinations by enabling abstention when uncertainty is high, this should be stated precisely, including a formal definition of the hallucination risk considered and the inherent limitations.

The narrative is difficult to follow. The paper mixes theory, system design, and empirical sections without a clear guiding thread. The Gram-geometry contribution feels detached from the CRC part; the motivation and relative value over simpler embedding-based scores need clarification. The novelty, beyond applying CRC with a different score and calibration variants, is moderate. Experiments are limited to short-form QA and do not cover more challenging generative settings, robustness under shift, or safety-critical tasks.

**Questions:**

- Can you clearly define what you mean by “hallucination” early in the paper, and explain how it differs from the type of risk controlled by CRC?

---

### Official Review · Reviewer_Vjr8 · 2025-10-30

**Soundness:** 2
**Presentation:** 2
**Contribution:** 2
**Rating:** 4
**Confidence:** 4

**Summary:**

This paper introduces a Conformal Actuator framework for controlling LLM randomness through Conformal Risk Control (CRC). The authors propose two calibrators (BB-CRC and RBWA-CRC) to reduce computational overhead while maintaining statistical validity, and present a Gram matrix geometry method for label-free uncertainty quantification. The framework is evaluated on factuality control and LLM-as-judge reliability across four QA datasets.

**Strengths:**

Innovative combination of Gram geometry with conformal risk control providing mathematically principled uncertainty quantification.

RBWA-CRC demonstrates superior calibration stability through unbiased smoothing and anti-concentration properties with single scalar actuator enabling efficient deployment.

Formal finite-sample guarantees under exchangeability assumptions with provider-agnostic design validated across multiple vendors.

**Weaknesses:**

Gram method only provides relative ranking within batches rather than absolute factuality assessment.

No mechanism ensuring high consensus corresponds to factual accuracy.

FS labels based on BERTScore similarity rather than ground truth factuality (Equation 5.1).

LLM-as-Judge creates circular dependency with systematic bias (Section 5.1, Appendix B.2).

Binary labels defined by generation method rather than actual factuality (Appendix A.2.2).

Gram geometry assumes meaningful embedding clustering without validation.

"Calibrate-once, deploy-often" paradigm presented as novel but is standard conformal prediction practice. Computing quantiles of non-conformity scores as thresholds then comparing test scores is established workflow since original conformal prediction literature.

**Questions:**

Can authors provide validation against fact-checking databases to demonstrate proxy metrics correlate with real factuality?

How do authors justify circular dependency in LLM-as-Judge evaluation?

What methods detect exchangeability assumption violations in practice?

How sensitive is Gram geometry to embedding quality and encoder choice?

How does performance scale beyond small experimental batch sizes (Table 4)?

Are there principled selection criteria or does this require extensive domain-specific validation?

---

### Official Review · Reviewer_A7gQ · 2025-10-31

**Soundness:** 3
**Presentation:** 1
**Contribution:** 2
**Rating:** 2
**Confidence:** 3

**Summary:**

This paper addresses hallucination control in LLMs by applying Conformal Risk Control (CRC). The authors propose two methods: Batched Bootstrap CRC and Randomized Batched Weighted-Average CRC, along with a Gram matrix-based uncertainty measure. The framework provides finite-sample risk guarantees for deciding whether to ship, abstain, or regenerate LLM outputs. Experiments across four QA datasets show factuality improvements when using either Gram-based consensus scores or LLM-as-judge scores as the policy signal.

**Strengths:**

1.	This paper addresses a crucial problem: controlling hallucinations and quantifying uncertainty in LLMs.
2.	The proposed framework provides finite-sample guarantees, building on CRC.
3.	The experimental scope is extensive and includes multiple datasets.

**Weaknesses:**

1.	This paper is very dense and not well organized.  It tries to cover too much: Gram geometry, two CRC variants, two policy scores, hallucination control, and LLM-as-judge, but the result is a dense paper that is hard to read and follow. The notations are not sufficiently motivated or well-defined.
2.	The proposed approach is not novel, as it combines existing techniques (CRC, bootstrap, Gram matrices)
3.	The paper cites recent hallucination detection works (SelfCheckGPT, semantic entropy probes) but doesn't compare against them experimentally.
4.	Theorem 3.1 relies on strong assumptions that are not sufficiently motivated.

**Questions:**

### Questions
1.	What are the actual acceptance rates at different $\alpha$ values?
2.	How does performance change with different embedding models?
### minor comments
1.	$\lambda_\text{max}$ appears in Algorithm 4.1 (line 6) without definition.

---

### Official Review · Reviewer_wo26 · 2025-11-02

**Soundness:** 3
**Presentation:** 2
**Contribution:** 3
**Rating:** 6
**Confidence:** 3

**Summary:**

This paper proposes a conformal actuator framework that applies Conformal Risk Control (CRC) to control LLM hallucinations and assess LLM-as-judge reliability. The authors introduce two computational improvements, Batched Bootstrap CRC (BB-CRC) and Randomized Batched Weighted-Average CRC (RBWA-CRC), that reduce calibration costs while maintaining finite-sample validity. They also present a Gram matrix-based semantic quantification method for uncertainty scoring. Experiments on four QA datasets demonstrate that the framework can reduce factuality errors at specified risk budgets, with the Gram-energy score showing more consistent gains than LLM-judge scores across settings.

**Strengths:**

1. The BB-CRC and RBWA-CRC variants provide theoretically grounded methods to reduce calibration costs (fewer LLM calls via bootstrapping/weighted averaging) while maintaining finite-sample validity, which is valuable for real deployment scenarios.

2. The evaluation spans four diverse QA datasets with meaningful ablations (entropy stress test, vendor swap), demonstrating robustness of the approach across different failure modes and implementation choices.

3. The framework cleanly separates calibration (which uses ground truth) from deployment (label-free), providing practitioners with a single interpretable threshold parameter that controls risk at a user-specified budget, which is operationally appealing.

**Weaknesses:**

1. The paper essentially combines existing ideas (CRC, Gram matrices for semantic uncertainty, bootstrap aggregation) rather than introducing fundamentally new methods. While the combination is useful, the individual components are well-established, and the specific contribution over prior CRC applications to LLMs (e.g., Yadkori et al. 2024) is incremental.

2. The entire framework relies on exchangeability for validity guarantees, yet realistic LLM deployments involve distribution shift from prompt drift, adversarial queries, and evolving user behavior. The authors acknowledge this limitation but provide no robustness analysis or empirical investigation of how the method degrades under violation—this gap significantly limits the practical applicability claims.

3. While experiments show QE beats QJ empirically, the paper lacks deeper investigation into the mechanisms. Is it robustness to outliers? Encoder quality? Task characteristics? Without this analysis, practitioners have limited guidance on when to choose which scoring function, and the Gram-geometry contribution feels under-explored.

4. The introduction motivates the work broadly (hallucination, prompt-injection, inconsistent evaluation), but experiments focus narrowly on QA factuality. Key questions remain unaddressed: Does the method work for open-ended generation? How does it handle multi-turn dialogue? What about safety risks beyond factuality? The generality claims are not sufficiently validated.

5. While BB-CRC/RBWA-CRC claim to reduce calls, the paper provides no quantitative comparison of total computational cost (calibration + deployment) versus baselines, nor wall-clock timing experiments. The batch size, embedding dimensionality, and number of responses per query are unspecified, making reproducibility and scalability assessment difficult.

**Questions:**

1. **How does the method degrade under distribution shift?** Can you provide empirical evidence on robustness when calibration and deployment distributions differ (e.g., calibrate on one dataset, deploy on another)? Even simple experiments would help assess practical viability and guide practitioners on when recalibration is needed.

2. **What drives the performance gap between QE and QJ?** Can you provide ablations isolating factors like embedding model choice, batch size, centering vs. non-centering, or task properties? Understanding when and why the Gram-energy score succeeds would strengthen the contribution and provide actionable insights for method selection.

---

### Note · Authors · 2025-12-03

I have read and agree with the venue's withdrawal policy on behalf of myself and my co-authors.